# Breaking the circularity in circular analyses: Simulations and formal treatment of the flattened average approach

Howard Bowman[1,2]*, Joseph L. Brooks[3], Omid Hajilou[1], Alexia Zoumpoulaki[4], Vladimir Litvak[5]

1 School of Computing, University of Kent, Kent, United Kingdom, 2 School of Psychology, University of Birmingham, Birmingham, United Kingdom, 3 School of Psychology, Keele University, United Kingdom, 4 School of Computer Science and Informatics, Cardiff University, Cardiff, United Kingdom, 5 Wellcome Centre for Human Neuroimaging, University College London, London, United Kingdom

* H.Bowman@kent.ac.uk

**Data Availability Statement:** Software can be found here: https://osf.io/4xnfc/. Code can be found here: https://osf.io/jntvf/.

**Funding:** The authors received no specific funding for this work.

## Abstract

There has been considerable debate and concern as to whether there is a replication crisis in the scientific literature. A likely cause of poor replication is the multiple comparisons problem. An important way in which this problem can manifest in the M/EEG context is through post hoc tailoring of analysis windows (a.k.a. regions-of-interest, ROIs) to landmarks in the collected data. Post hoc tailoring of ROIs is used because it allows researchers to adapt to inter-experiment variability and discover novel differences that fall outside of windows defined by prior precedent, thereby reducing Type II errors. However, this approach can dramatically inflate Type I error rates. One way to avoid this problem is to tailor windows according to a contrast that is orthogonal (strictly parametrically orthogonal) to the contrast being tested. A key approach of this kind is to identify windows on a *fully flattened* average. On the basis of simulations, this approach has been argued to be safe for post hoc tailoring of analysis windows under many conditions. Here, we present further simulations and mathematical proofs to show exactly why the Fully Flattened Average approach is unbiased, providing a formal grounding to the approach, clarifying the limits of its applicability and resolving published misconceptions about the method. We also provide a statistical power analysis, which shows that, in specific contexts, the fully flattened average approach provides higher statistical power than Fieldtrip cluster inference. This suggests that the Fully Flattened Average approach will enable researchers to identify more effects from their data without incurring an inflation of the false positive rate.

## Author summary

It is clear from recent replicability studies that the replication rate in psychology and cognitive neuroscience is not high. One reason for this is that the noise in high dimensional neuroimaging data sets can "look-like" signal. A classic manifestation would be selecting a region in the data volume where an effect is biggest and then specifically reporting results

**Competing interests:** The authors have declared that no competing interests exist.

on that region. There is a key trade-off in the selection of such regions of interest: liberal selection will inflate false positive rates, but conservative selection (e.g. strictly on the basis of prior precedent in the literature) can reduce statistical power, causing real effects to be missed. We propose a means to reconcile these two possibilities, by which regions of interest can be tailored to the pattern in the collected data, while not inflating false-positive rates. This is based upon generating what we call the Flattened Average. Critically, we validate the correctness of this method both in (ground-truth) simulations and with formal mathematical proofs. Given the replication "crisis", there may be no more important issue in psychology and cognitive neuroscience than improving the application of methods. This paper makes a valuable contribution to this improvement.

This is a *PLOS Computational Biology* Methods paper.

## Introduction

A number of papers in cognitive neuroscience or related disciplines have questioned the reliability of the statistical methods and practices being employed, and their consequences for the replicability of findings in the published literature [1–10] (where replicability is used to mean a study that arrives at the same finding as a previous study through the collection of new data, but using the same methods as the first study). In one way or another, these articles are highlighting difficulties associated with handling the multiple comparisons problem, whether in the implementation of the methods employed or the practices of experimentalists [5,8]. The latter of these (experimental practice) may be particularly pernicious, since it rests upon research team practices that are unlikely to be reported in an article. For example, if a laboratory routinely tries various pre-processing settings, but only reports the analysis that yielded the smallest p-value, it is very hard to assess the reliability of a finding unless one can somehow count the number of settings tried (actually, it is even difficult to do this accurately when you know the number of settings tried, since different settings will be somewhat correlated).

In response to this, many have argued for systematic procedures that force scientists to pre-specify the settings (or more formally the hyper-parameters) of their analyses (such as pre-processing settings), before starting to collect data. A prominent proposal is registered reports, e.g. [12], whereby a journal accepts to publish a paper on the basis of a prior statement of the experiment, its methods, materials and procedures, whether a significant result is eventually found or not. For neuroimaging studies, this may include specifying the region-of-interest (ROI) where effects are going to be tested for in the data (e.g. electrodes and time periods). This is an excellent strategy for controlling the false positive rate in the literature, and will surely increase the replicability of published studies. However, some naïve approaches to pre-registration have limitations, especially in the context of complex neuroimaging data sets.

In particular, within Event Related Potential (ERP) research, it is often difficult to know exactly where in space (i.e. electrodes) and time an effect will arise, even if one has a good idea from previous literature of the ERP component that responds to the manipulation in question. Small changes in experimental procedures, or of participant group, can have a dramatic effect on the latency, scalp topography and, even, the form of a component. For example, Fig 1 shows ERP grand averages from two studies that used very similar stimulus presentation procedures and timing; see Supporting Information S5 Text for details. Certainly, the upper panel

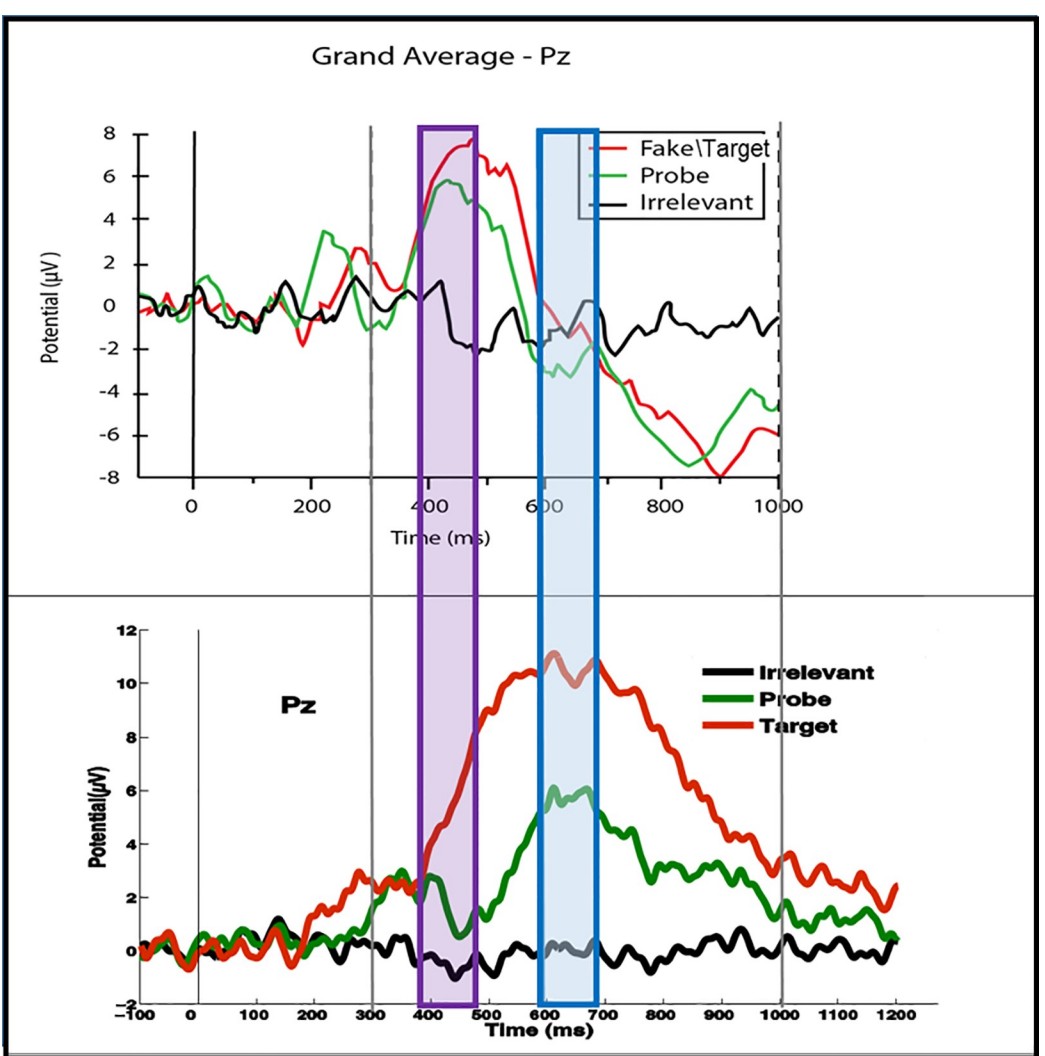

**Fig 1. ERPs from two Rapid Serial Visual Presentation (RSVP) experiments at the Pz electrode.** The top panel experiment was published in [11]. The lower panel shows unpublished data. The experiments use very similar presentation paradigms, with name stimuli in both cases; see Supporting Information S5 Text for details. Even though the design differences between the two experiments are small, the timing and form of the P3b component is very different. Of particular interest here is that the Target P3bs (red lines) were very different in the two experiments, as were the Probe P3bs (green lines). For example, the blue region marks the peak of the Probe P3b in the second experiment (lower panel), which misses the corresponding Probe P3b peak in the first experiment (upper panel). In fact, the misalignment of the P3b effects in the two experiments is so great that the P3b in the second experiment is aligned with the negative rebound to the P3b peak in the first experiment. Additionally, the purple region marks the peak of the Probe P3b in the upper panel, which clearly precedes the peak in the lower panel.

experiment was as good a precedent for the lower panel experiment (which came later) as could be found within the literature or the trajectory of the research programme of which they were a part [11,13]. Despite the similarity between the experimental paradigms, the timing and form of the P3 components are very different. This can, for example, be seen with the Probe condition (the green time series), where the P3 peak in the lower panel actually arises approximately 200 ms later, during the negative rebound phase of the P3 in the upper panel; see blue region. There are many potential reasons for these differences, some of which are discussed in Supporting Information S5 Text. However, critically for this paper, the ERP landmarks (e.g.

peaks) are very different in these two closely related experiments. This is a particularly compelling demonstration of the problems of using prior precedent to define an ROI in ERP analysis, since the data sets for both these experiments were collected by the same team with the same basic pre-processing and analysis methods. A change in team, which is the norm when comparing studies in the literature, should only make the disparity between ERPs greater. Additionally, although we have focussed on misalignment in time, a prior precedent may also misalign in space, i.e. on the scalp.

While pre-registration is a highly important response to the replicability crisis, if one is limited to using previous studies for defining *fixed position* regions-of-interest (i.e. using prior precedent) within the pre-registration approach, the Type II error rate (i.e. missed effects) may increase and make it more difficult to detect novel effects or effects that are subject to significant inter-experiment variation (this is the standard trade-off between type I and type II error rates, e.g. one could effectively make the threshold for judging significance more stringent, but this will increase the probability that real effects will be missed). The opportunity to report exploratory analyses within the pre-registration framework clearly helps with this problem. For example, one could perform an exploratory whole-volume analysis. However, such a finding is likely to have less statistical power than an ROI analysis (see section "**Statistical Power**" for a demonstration of this) and would, by virtue of being labelled exploratory, not have the same status as a successfully demonstrated pre-registered finding.

One approach to overcoming the limitations of a priori ROI selection is to use a data driven method, which uses features of the collected data to place the ROI. Although data driven approaches may, at first consideration, seem incompatible with pre-registration, if the method and properties of the approach are chosen in advance of the study then it can be performed without inflating the Type I error rate, e.g. [8].

An elegant way to do this is via a contrast that is orthogonal to the contrast of the effect of interest, e.g.[14,8]. Thus, a first *selection* contrast is applied to identify the region at which to place the analysis window, and then a distinct *test* contrast is applied at that region. As long as these two contrasts are, in a very specific sense, orthogonal (in fact, parametrically contrast orthogonal–see the mathematical formulation later in this paper), they will have the property that for null data, there will be no increased probability of the test contrast being found significant in a window/ROI determined by the selection contrast, than in any other region not selected (In fact, a stronger property would hold, viz that the distribution of possible p-values for the test contrast under the null hypothesis is uniform). The logic here then is that comparisons can be accumulated, as long as they are *not* accumulated with regard to the effect being tested.

Brooks et al [8] proposed a particularly simple orthogonal contrast approach, called the *aggregated average*. A central concern of the current paper is to explain why this approach does not inflate the type-I error rate. With classical frequentist statistics, maintaining the false positive rate of a statistical method at the alpha level ensures the soundness of the method. Statistical power (one minus the type II error rate) is, of course, also important; that is, we would like a sensitive statistical procedure that does identify significant results, when effects are present.

Brooks et al [8] provided a simulation indicating that the aggregated average approach to window selection is more sensitive than a fixed-window prior precedent approach when there is latency variation of the relevant component across experiments. This is, in fact, an obvious finding: with a (fixed-window) prior precedent approach, the analysis window cannot adjust to the presentation of a component in the data, but it can for the aggregated average.

A more challenging test of the aggregated average's statistical power is against mass-univariate approaches, such as, the parametric approach based on random field theory implemented

in the SPM toolbox [15] or the permutation-based non-parametric approach implemented in the Fieldtrip toolbox [16–17]. This is because such approaches do adjust the region in the analysis volume that is identified as signal, according to where it happens to be present in a data set. However, because mass-univariate analyses familywise error correct for the entire analysis volume, their capacity to identify a particular region as significant reduces as the volume becomes larger. In contrast, the aggregated average approach is not sensitive to volume size in this way, implying that it could provide increased statistical power, particularly when the volume is large. One contribution of this paper, is to confirm this intuition in simulation; see section "**Statistical Power**".

However, there are subtleties to the correct application of the aggregated average approach and the orthogonal contrast method in general. A thoughtful presentation of potential pitfalls can be found in the supplementary material of [5]. As reported there, showing that the contrast vectors for Region-Of-Interest (ROI) selection and test are orthogonal is not sufficient to ensure orthogonality of the results of applying the contrasts, with a particular experimental design (i.e. design matrix) and data set (note, for the fully flattened average method we are advocating it will actually not even be necessary). Kriegeskorte et al argued that three properties need to hold to ensure the false positive rate is not inflated. These are, 1) *contrast vector orthogonality*: ROI selection and test contrast vectors need to be orthogonal (i.e. the dot product of the vectors is zero), 2) *balanced design*: the experimental design (i.e., design matrix) needs to be balanced (e.g. trial counts should not be different across conditions), and 3) *absence of temporal correlations*: temporal correlations should not exist between the data samples to be modelled. The second of these is important, since different trial counts between conditions can arise for many reasons, such as artefact rejection or since condition membership is defined by behaviour (e.g. whether responses are correct or incorrect). With careful experimental design, the third of these (temporal correlations) can be avoided in many M/EEG studies (for clarification of this point see Supporting Information S1 Text). However, dependences across trials/ replications can sometimes arise, such as from very low frequency (across trial) components (e.g. the Contingent Negative Variation [18]) or learning effects across the timecourse of an experiment. We will return to these three proposed safety properties (contrast vector *orthogonality*, *balanced design* and *absence of temporal correlations*) a number of times during this article.

Our objective here is to further characterise, demonstrate the validity and statistical power of, and show the generality of a simple orthogonal contrast approach that we recently introduced [8], which we named the Aggregated Grand Average of Trials (AGAT). The treatment of this issue here is more general than in [8], in the sense that we accommodate analyses in which the random effect (i.e. unit over which inference is performed) could be trials, items, participants, etc. The problem that we are seeking to resolve arises for all these different varieties of random effect; see the "**Discussion**" section for further details. Accordingly, in this paper, we call the orthogonal contrast approach we are advocating, the *Fully Flattened Average*, to capture the generality of our focus. Software implementing this orthogonal contrast approach is available at, https://sites.google.com/view/brookslab/downloadsresourcesstimuli/agat-method.

To fulfil the objectives of this paper, we will first review the Fully Flattened Average (FuFA) approach in section **"Background"**. Then, in section **"Unbalanced Designs–Simulations"**, we will investigate in simulation, what seems at first sight to be an oddity of the Fully Flattened Average approach in the context of unbalanced designs. This is the fact that simple averaging would cause the condition with fewer replications to have more extreme amplitudes than the condition with more replications (since noise is reduced through averaging). Of itself, such averaging would bias differences of peak amplitudes (or differences of mean amplitudes in

maximum windows) across unbalanced conditions and inflate false-positive rates. We will show in simulations why this averaging bias does not in fact inflate false positives for the FuFA appraoch, because there is effectively a second bias that works in perfect opposition to this bias due to averaging. Furthermore, we will show that this perfect opposition of the two biases does not obtain for the most obvious, and often used, means to obtain an aggregated average, which we call the Average with Intermediate Averages (AwIA) approach (see section **"Unbalanced Designs–Simulations"**). Thus, we show that overall, when both biases are considered, FuFA is not biased, but AwIA is. Following this, in section **"Temporal Correlations–Simulations"**, we present simulations that suggest that these bias freeness properties generalise to data sets with temporal correlations across replications. We then give formal background to the new Fully Flattened Average (FuFA) method and the properties it should satisfy (see section"**Why the FuFA is Unbiased–Formal Treatment"**), before presenting a formal mathematical treatment of the FuFA and AwIA methods. This will enable us to verify mathematically that the FuFA is not biased under reasonable assumptions (see section"**Why the FuFA is Unbiased–Formal Treatment"**), providing a fully general verification of the method, compared to the more limited scope of the simulations. This will show that an orthogonal contrast approach does not need to meet the balanced design assumption. Finally, in section **"Statistical Power"**, we will also show that the FuFA approach can increase statistical power over cluster-based family-wise error correction, the de-facto standard data-driven statistical inference procedure employed in neuroimaging.

## Background

### Aggregated averages

If we assume a simple statistical test, such as a t-test, is to be performed between two conditions in an M/EEG experiment (or other spatiotemporal dataset), then perhaps the simplest attempt at an orthogonal contrast is to just collapse across the two conditions by averaging waveforms. Assuming that the waveforms have similar features and similar latencies of features, this will produce an average with any landmark (e.g. a peak) that is common to the two conditions still present. Importantly, under the null hypothesis, large differences between conditions should be as likely to occur at any position in the data, with pure sampling error determining whether those differences do or do not fall at key common landmarks, such as peaks. We call the resulting time-series an *Aggregated Average* due to the aggregation of data across conditions. One can then select windows/ regions of interest on this aggregated average, without, it is hoped, biasing (i.e., inflating the Type I error rate for) the t-contrast of interest under the null hypothesis [8].

There is, though, an important subtlety to how this aggregated average is constructed. Specifically, we differentiate two aggregation procedures, which are shown in Fig 2. The first involves a hierarchy of averaging, as would be performed in a classic ERP processing pipeline, producing what could be called, the *Average with Intermediate Averages* (AwIA). This involves averaging replications (e.g. trials/epochs) within each condition to form condition averages and then averaging condition averages to produce the AwIA. (Of course, experiments with further levels of hierarchy, e.g. trials, then participants, then conditions would involve a further level of intermediate averages in the AwIA approach.) In contrast, the second of these procedures aggregates at the replications level, flattening the averaging hierarchy to one level (although an alternative to flattening is to take weighted averages, as we will elaborate on later). An aggregated average is then generated from this flattened set, producing what could be called the *Fully Flattened Average* (FuFA).

Importantly, the AwIA and FuFA are only the same if replication counts are equal across conditions, i.e. in balanced-design experiments. As we will justify in simulation and proof, it

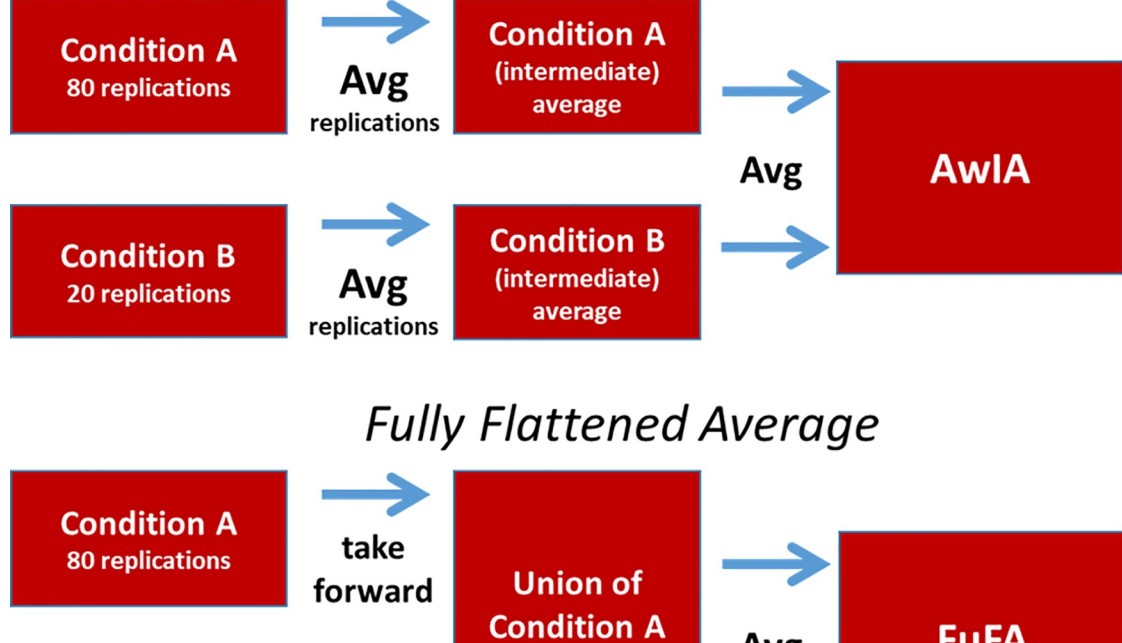

**Fig 2. Two possible methods for generating an aggregated average.**

turns out that only the FuFA is unbiased for use in selecting regions-of-interest, i.e. does not inflate the false positive rate, in the presence of an unbalanced design.

## Notation

Although we defer our formal treatment of orthogonality of contrasts until section **"Why the FuFA is Unbiased–Formal Treatment"**, to frame our discussion, we present some basic General Linear Model (GLM) notation here. We focus on the two-sample (independent) t-test case. Using the terminology in [15,19], we define $c_t$ to be the t-test contrast vector, i.e.

$$c_t = [+1, -1]$$

and $X$ denotes the standard two-sample t-test design matrix, i.e.

$$X = \begin{pmatrix} 1 & 0 \\ \vdots & \vdots \\ 1 & 0 \\ 0 & 1 \\ \vdots & \vdots \\ 0 & 1 \end{pmatrix}$$

where the first column is the indicator variable for condition 1 and the second for condition 2. $X$ defines that we have two conditions, and $c_t$ that we seek to test the difference of means of these conditions. The dependent variable (i.e. the data) would be expressed here as a (column) vector of samples that run down the entire course of the experiment. For example, these could be all the samples of a particular time-space point, e.g. a time relative to stimulus onset and a particular electrode in space, in a mass-univariate analysis [15,19]. Alternatively, samples could be mean amplitudes across intervals of a particular size, e.g. average amplitude in a 100ms window, as is common in the traditional ERP approach [20]. The resulting data vector, denoted $y$, runs across all conditions.

Unbalanced conditions could result, for example, from replication count asymmetry. For example, the following design matrix indicates three data samples in condition 1 and four in condition 2.

$$X = \begin{pmatrix} 1 & 0 \\ 1 & 0 \\ 1 & 0 \\ 0 & 1 \\ 0 & 1 \\ 0 & 1 \\ 0 & 1 \end{pmatrix}$$

Given such a design matrix, the simplest ROI selection contrast that one could apply would correspond to the contrast vector,

$$c_{s,AwIA} = c_{s,IA} = [1/2, 1/2]$$

This is the AwIA contrast under the standard processing pathway; that is, the ROI is selected using the average of the averages of the two conditions.

We can also formulate the FuFA in this setting. Consider the design matrix $X$ above. Under the $c_{s,IA}$ contrast, data-samples associated with the first condition (the smaller one) contribute more to the aggregated average than those from the second. In contrast, in the FuFA, all data-samples contribute equally to the aggregated average. Such equality of contribution can be obtained in the GLM setting by simply taking a weighted average, when building the aggregated average from its condition averages. Accordingly, we define the FuFA selection contrast vector as,

$$c_{s,FuFA} = c_{s,FA} = [N_1/N, N_2/N]$$

where $N_1$ is the number of data-samples in condition 1 (i.e. 1's in the first column of the design matrix) and $N_2$ the number of data-samples in condition 2, while $N = N_1+N_2$ (the number of rows in the design matrix). In this contrast, the smaller condition is down weighted, relative to the bigger one, ensuring that each replication (whether in the larger or smaller condition) contributes equally to the aggregated average.

How then do the previously discussed candidate safety properties, arising from [5] manifest in this GLM model?

1. *Contrast vector orthogonality*: this would hold, if the dot product of the selection and test vectors was zero.

2. *Balanced design*: as previously discussed, this would hold if the design matrix was balanced, i.e. $N_1 = N_2$ in the above illustration.

3. *Absence of temporal correlations*: this would hold if the data, which would become the dependent variable in the GLM regression, contained no correlations down its time-course; this amounts to there being no "carry-over" effects from sample-to-sample, i.e. between replications in an M/EEG experiment.

With regard to these properties, $c_t$ and $c_{s,IA}$ are indeed orthogonal (the dot product of the vectors is zero), however, $c_t$ and $c_{s,FA}$ are in fact not orthogonal. That is, in terms of our earlier example, the following holds,

$$c_t . c_{s,IA}^T = [+1, -1].[1/2, 1/2]^T = 0$$

and,

$$c_t . c_{s,FA}^T = [+1, -1].[3/7, 4/7]^T = -1/7.$$

We will return to this issue of contrast vector orthogonality in section **"Why the FuFA is Unbiased–Formal Treatment"**.

With regard to temporal correlations, with careful experimental designs, in most cases in the M/EEG context, temporal correlations across data samples (which are replications/trials in M/EEG) can be avoided (see Supporting Information, S1 Text, for further discussion on this issue). However, as previously discussed, such structure in replications can arise in particular experimental contexts. Accordingly, we include a consideration of the consequences of temporal correlations across replications, at least partly to inform Kriegeskorte et al's discussion of this issue; see Supporting Information S6 Text.

## Unbalanced designs–simulations

**Statistical bias.** We are interested in identifying statistical bias, with the term used in the standard statistical sense, induced by procedures for selecting regions-of-interest in M/EEG studies. Specifically, a bias exists if the estimate of a statistic arising from a statistical procedure is systematically different to the population measure being estimated. For us, the measure of interest will be the difference of mean amplitudes in an ROI between two conditions, where the key point for this paper is how these ROIs are identified.

This paper discusses statistical power in section "**Statistical Power**", but its main focus is on false positive (i.e. type I error) rates. In our false positive simulations, in a statistical sense, the difference of mean amplitudes in a selected ROI measure will be, by construction, zero at the population level, since the null hypothesis will hold. We will, then, be assessing the extent to which two distinct methods for identifying regions-of interest (according to maximum mean amplitudes) create a tendency across many simulated null experiments for the mean amplitude for one condition to be larger than the mean amplitude for the other. If a given method does this, then the method has a bias. This is because the selection of the ROI will be consistently associated with a difference between conditions that is (in a statistical sense) different from zero. This would not arise from an unbiased procedure under the null hypothesis.

In our previous work [8], we have directly assessed false positive rates, by running statistical tests on each simulated data set and then counting up the number of p-values that end up below the critical alpha level, which is typically 0.05, e.g. Fig 2 in [8]. Each such data set with a significant p-value is a false-positive, and in the limit, if the method is functioning correctly, the percentage of such false-positives should be 100 x alpha (i.e. typically 5%). Identification of

a bias of the kind discussed above would be expected to induce an inflation or deflation, of the rate of false positives (making it different to 5%).

**An oddity.** A key aspect of the FuFA approach is that (unlike the AwIA) it is bias-free for unbalanced designs. This might, at first sight, seem surprising because, in unbalanced designs, the simple averaging associated with generating condition averages will induce an amplitude bias between the Small (i.e. fewer replications) and Large conditions. That is, the average waveform in the Large condition will have less extreme amplitudes generated by noise, than that of the smaller condition.

This difference in extreme values will, in turn, introduce a tendency towards differences between the conditions that are (in a statistical sense) different from zero. Condition differences that are (statistically speaking) above zero under the null would translate into a higher Type I error rate. We call this the *Simple Averaging Bias*; see Fig 3 for an example. However, despite this bias at one point in the FuFA process, overall ROI selection using the FuFA does not inflate the Type I error rate. To somewhat pre-empt our findings, this is because there are in a sense two biases, which in the case of the FuFA, counteract each other, but in the case of the AwIA accumulate.

The second bias arises because the FuFA itself is more like the condition with more data samples (i.e. large condition) than the condition with fewer (i.e. small condition). Indeed, it even becomes almost identical to the Large condition when the asymmetry is big. This can be seen, for example, in Fig 4, particularly Panel B, where the FuFA subpanel (b), is almost identical to the Large condition average, subpanel (f). Accordingly, the window selection performed

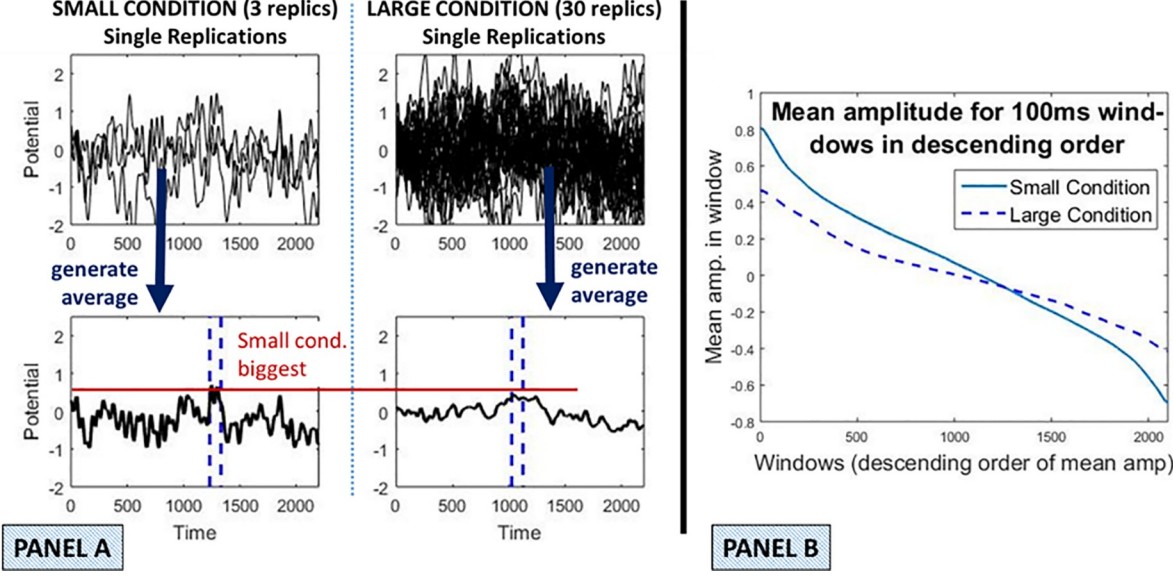

**Fig 3. Illustration of (simple) averaging bias.** Two conditions with different replication counts were generated according to the properties introduced in section "Simulations". The Small condition has three replications and the Large 30. A deliberately large asymmetry is considered for clarity of illustration. **Panel A**: Single replications are depicted overlaid in the upper two subpanels. Averages for these two conditions are depicted in the lower two subpanels. As would be expected, the Small condition average exhibits more noise and thus, more extreme values than the Large condition average. Accordingly, its highest mean amplitude is higher than for the Large condition, as illustrated with the red horizontal line. The blue dashed vertical lines indicate the highest amplitude 100 ms interval. **Panel B**: The property illustrated in Panel A that more averaging reduces extreme values, both highest (most positive) and lowest (most negative) amplitude, is illustrated more generally. The simulation of Panel A was run 100 times. In each simulation, we calculated the mean activity in a 100ms window at all possible locations at which the window could be placed on the average. We did this separately for the Small and Large conditions. Within each condition, we then sorted the window means from highest (leftmost) to lowest (rightmost) in panel B. This vector of highest to lowest mean amplitudes was then averaged across the 100 simulations, to obtain a (central tendency) estimate of the sequence of mean amplitudes in descending order. This was done for both Large and Small conditions and plotted in Panel B.

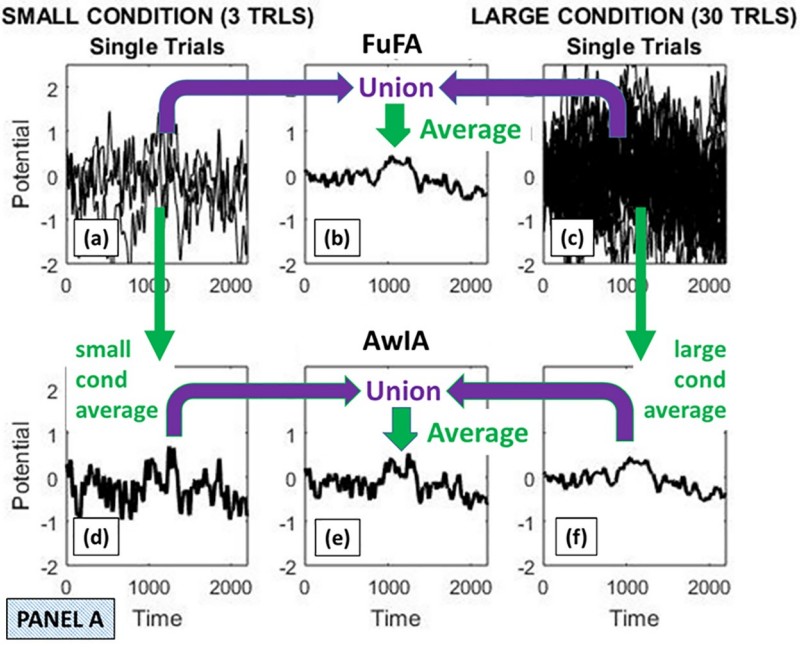

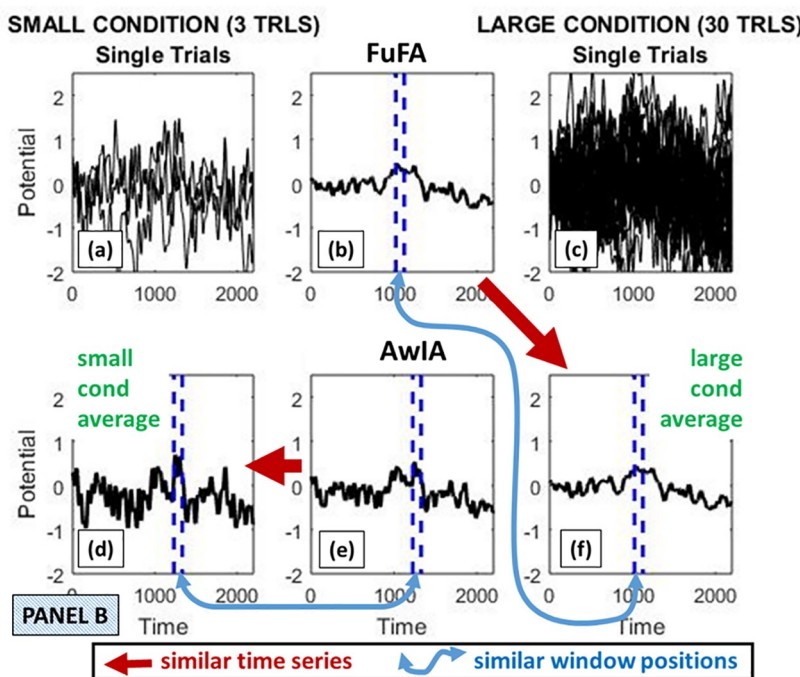

**Fig 4. Illustration of bias due to window selection using the same simulation run as in Panel A of Fig 3.** The top panel of this figure (Panel A), depicts how the FuFA and AwIA are generated. That is, the FuFA is an average of the union of all the replications from the two conditions. In contrast, the AwIA is an average of two time-series: the average of the Small condition and the average of the Large condition. The union in this case would contain two time-series, which are then averaged. Panel B shows that the FuFA and AwIA procedures generate very different time-series. Specifically, the key landmarks (e.g., maximum/minimum points) of the AwIA tend to correspond to those of the Small condition average. This is because the Small condition average has more extreme amplitudes, due to the (simple) averaging bias, so they "swamp" the less extreme amplitudes of the Large condition average, when the AwIA is generated. In contrast, the FuFA tends to be more like the Large condition average, since all single-replications contribute equally to it and there are more single-replications in the Large condition. This tendency can be seen in the

window placements. Windows are placed in the FuFA, AwIA, average Small and average Large, with, in each case, the 100ms window with the highest mean amplitude selected, and shown by the blue dashed vertical lines. The AwIA window ends up at a similar position to in the Small condition average, while the FuFA window ends up at a similar position to in the Large condition average.

on the FuFA will be biased towards the Large condition (i.e. with more replications). That is, it will, in a statistical sense (i.e. across many samplings), identify a window that is closer to the true maximum window placement of the Larger than of the Smaller condition. This observation that the FuFA is more like the large than the small condition stands against the belief that just by taking an average weighted by the proportion of contributing trials will generate an aggregated average in which the two conditions are equally represented. It is more complicated than that and best thought of as two counteracting biases. That is, these two biases, which we will call, the *simple averaging* bias and the *window selection* bias, act in opposite directions in the FuFA and thereby counter-act each other.

We will first illustrate this notion that there are two biases (see section "**Two Biases**") and then confirm this with a null hypothesis simulation of the two methods (see section "**Simulations of FuFA and AwIA**"). In this way, our simulations will clarify why the bias introduced by simple averaging does not generate an overall bias in the FuFA approach.

## Construction of simulations

We present null hypothesis simulations of the FuFA and AwIA, while varying the replication count asymmetry between two conditions. The simulations have the following main characteristics.

- replication time-series comprise 2200 time points;

- the same signal was included in every replication time-series;

- (coloured) noise time series were overlaid on top of the signal; these noise time series were generated according to the human temporal frequency spectrum, using the algorithm devised by [21], which was employed in [8] and in [22], we give details in Supporting Information S7 Text;

- each simulated data set comprised two conditions, which we call Small and Large according to the number of replications;

- in all cases, the null hypothesis held; that is, the replications in the two conditions were in a statistical sense, the same, i.e. were drawn from the same distribution, with the only difference being due to sampling variability of noise;

- in section "**Two Biases**", we use an integration window of 100ms width for illustrative purposes (i.e. our dependent measure is average amplitude across a 100 ms window), but then in the full simulation in section "**Simulations of FuFA and AwIA**", *peak* amplitude will be taken as the dependent measure, i.e. an integration window of size one was employed (such a narrow window was used, since our earlier simulations [8] have shown that the greatest bias with unsound methods can be observed for single time-point windows, making it an appropriate test of bias freeness); and

- in the full simulations, we ran the two aggregated average methods on the peak.

### Two biases

As previously discussed, there are two distinct ways in which an unbalanced (i.e. more data in one condition than another) design has a differential effect on the inference process. We call these:

1. (simple) *averaging bias*, and

2. *window selection bias*.

We discuss these in turn.

**Simple Averaging Bias.**    The *averaging bias* is independent of whether a FuFA or AwIA is used, and arises simply because extreme amplitudes reduce when more replications contribute to an average. This is illustrated in Fig 3, where we compare the averages generated from a Small and a Large condition. The null-hypothesis holds, since, as just discussed, the same signal is included in both conditions, and noise with the same properties, is overlaid on both. The only difference, in a statistical sense, between the two conditions is the number of replication time-series they comprise.

As can be seen in Fig 3, averaging reduces extreme values; indeed, this is the logic of the Event Related Potential (ERP) method in the first place–noise is averaged out, revealing the underlying signal. This is particularly clear in Panel B of Fig 3, where mean amplitudes in 100ms windows are more extreme in the Small condition, apart, of course, at the point of cross-over. Accordingly, the difference in mean amplitudes in maximal windows between Small and Large conditions will be biased: in general, the max window mean amplitudes of Small will be higher than for Large, even though the null hypothesis holds by construction. Importantly, because the aggregated average processes (both FuFA and AwIA) select the highest amplitude windows in the aggregated grand average (or lowest amplitude for negative polarity components), they will be biased (in this averaging sense) and the condition with fewer replications will (in a statistical sense) have higher amplitudes.

To be clear, the aggregated average methods will not typically select the highest window in either Small or Large conditions, since the form of these aggregated averages is influenced by both conditions, however, it will tend to select a window that is high amplitude in both conditions (since the aggregated average is comprised from them). In this sense, the aggregated average methods will tend to select windows in the component conditions that are high amplitude amongst the possible windows, and, all else equal, these will tend to be higher in the Small condition than in the large condition.

**Window selection bias.**    The *window selection bias* arises, since the aggregated averages are differentially impacted by the constituent conditions according to their replication count. This is illustrated in Fig 4, where (the top) Panel A shows how the AwIA and FuFA are generated, and (the bottom) Panel B shows the selection bias. That is, the FuFA is more like the average of the Large condition, while the key (extreme value) landmarks of the AwIA are more like those of the Small condition. This is reflected in the placement of the maximum 100ms mean amplitude windows on each waveform in Panel B. The selected maximum window in the FuFA is in a very similar position to that in the Large condition average, while the window in the AwIA is in a similar position to that in the Small condition. In this sense, FuFA window selection tends to bias towards the Large condition, while the AwIA window selection biases towards the Small condition. These would indeed create biases, since in either case, AwIA and FuFA, a tendency will be generated for one condition to have a mean amplitude in the selected window that is closer to that of its true max window than it is for the other condition. If all else were equal, this would create a bias towards the condition with window closer to its true max, yielding a higher mean amplitude. As a result, the difference of selected mean amplitudes would be (statistically speaking) different to zero under the null hypothesis.

Critically, as previously stated, the (simple) averaging bias and the window selection bias work in the same direction, and thus, accumulate, for the AwIA: they both bias towards the Small condition. That is, in a statistical sense, a window will be selected closer to the true maximum window placement of the Small condition, which, additionally, intrinsically has more

extreme values than the Large condition (note, this observation is somewhat inconsistent with guidance previously given in the literature, see Supporting Information S2 Text).

In contrast, and also as previously stated, the averaging and window selection biases work in opposite directions for the FuFA: (simple) averaging biases towards the Small condition, but window selection biases towards the Large condition. In addition, the biases are driven by the same across condition ratio of data-samples, are hence, equal and opposite, and accordingly, cancel.

## Simulations of FuFA and AwIA

To confirm this intuition, we present null hypothesis simulations of the FuFA and AwIA, while varying the replication count asymmetry between the two conditions. The simulations have the properties outlined in section "Construction of Simulations" with the following additional characteristics.

- each time point is an 8x8 spatial grid (corresponding to 64 sensors);

- a signal time-series was placed at each sensor of the central 2x2 region of the overall 8x8 grid;

- (coloured) noise time series of the kind outlined in section "**Construction of Simulations**" were overlaid at each point in the grid;

- spatial smoothing with a Gaussian kernel (of width 0.5) was applied on the grid at each time point;

- each simulated data set comprised 100 replications, divided into two conditions–Small and Large–according to the following asymmetries: 10/90, 20/80, 30/70, 40/60, 50/50;

- we determine the amplitude at the time-space-point (i.e. point in time by electrode volume) selected from FuFA or AwIA in the average of the *Small* and of the *Large*, i.e. our regions of interest are peaks.

Data generated from this simulation are shown in Fig 5, both a single replication (on left) and an average from 30 replications (on right). As would be expected, the common signal across replications emerges through averaging, with reduction of noise amplitudes.

The results of these simulations are shown in Fig 6. This shows clearly that the AwIA is biased by replication-count asymmetry. For example, in panel A, the amplitudes at the AwIA peak are bigger for the Small than the Large condition (see solid lines), so, the difference of the two (red vertical arrow) will be non-zero. In addition, this bias systematically reduces as replication-counts come into balance, i.e. as one moves from left to right in panel A.

As previously discussed, and elaborated on in the caption of Fig 6, the simple averaging bias (green arrow) and the window selection bias (purple minus blue arrows) accumulate for the AwIA, see Panel A, generating a substantial overall bias (red arrow) at big replication-count asymmetries. This is summarised in Panel B. (See Supporting Information S3 Text for a clarification of how these findings relate to those in Brooks et al [8].)

In contrast, the FuFA is free from bias at all asymmetries. This is summarised in Panel D, where it is evident that the averaging bias (which is the same for both FuFA and AwIA), is (perfectly) counteracted by the window selection bias. Accordingly, save for sampling error, the Overall Bias (the Red line) is zero at all asymmetries.

Interestingly, it is not just that the amplitudes at the FuFA peak are equal (i.e. the Overall Bias is zero), but those amplitudes are constant across replication asymmetries. In other words, it is not just that the solid lines in panel (C) of Fig 6 are equal across all replication-

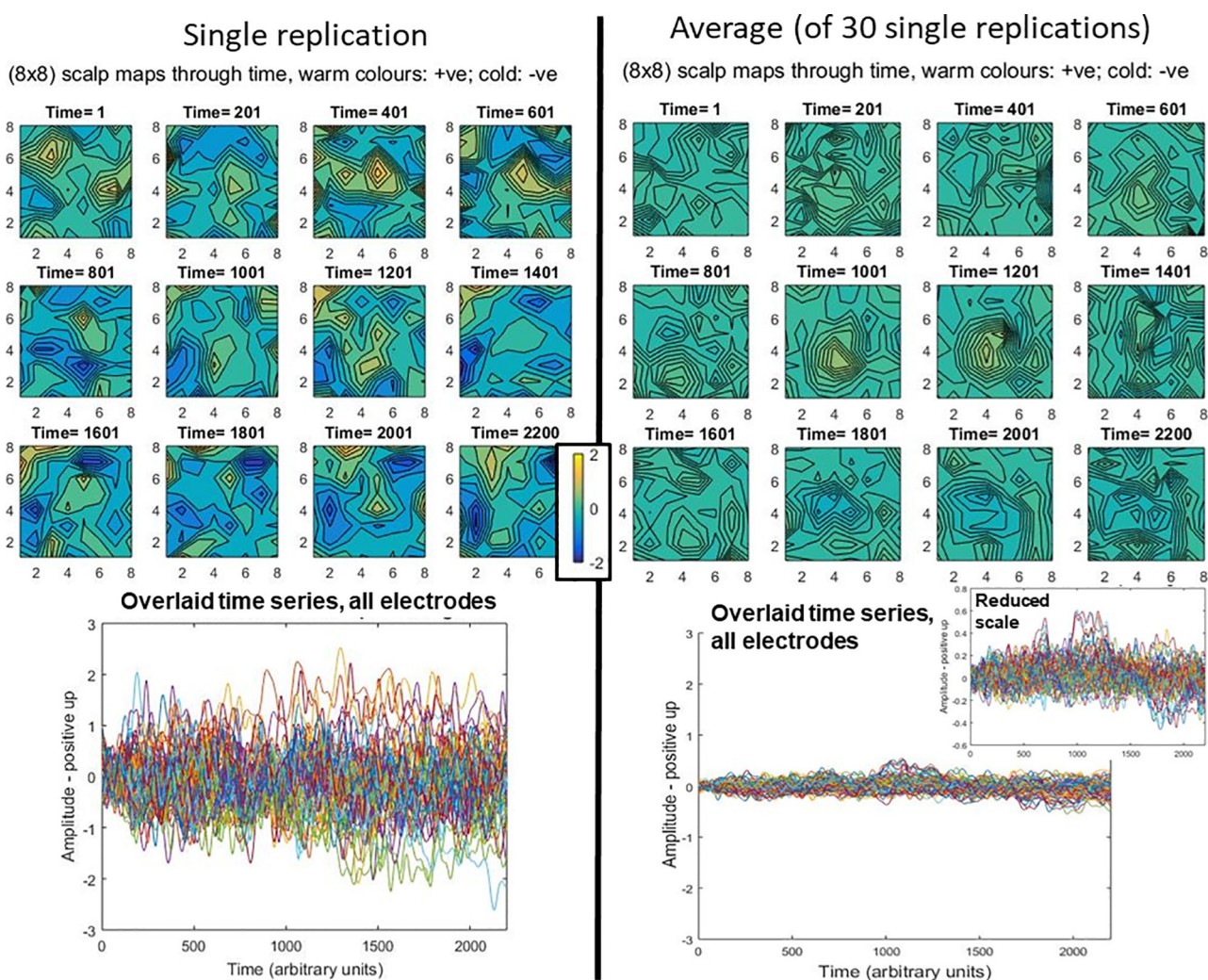

**Fig 5. Illustrative data generated under null-hypothesis simulation.** The left side shows a typical single replication, while the right side shows a typical average, here generated from 30 replications. In both cases, we present the same data in two different ways. First, (at the top) scalp topographies through time are presented, with the two topography sequences using the same colour scale to aid comparison. Second, (at the bottom) the time-series at each electrode are presented overlaid in the same plot. The two main plots have the same scale to aid comparison between amplitudes of single replication and average. Consistent with the averaging bias, the single replication contains much more extreme deflections (both positively and negatively). This can be seen in the more extreme colours in the left-hand scalp topographies, and the larger amplitudes in the left-hand overlaid time-series plot. The reduction in extreme amplitudes evident on the right side due to averaging, has enabled the signal to emerge. This can be seen as a positive deflection at the centre of the grid, at time-points 1001 and 1201, and a negative one also at the centre of the grid, in the time-range 1801–2200. As would be expected, the overlaid time-series plot of the average shows the signal landmarks in the same time periods, see particularly, inset plot on the right.

count asymmetries, but they are also *horizontal*. There is, then, a sense to which there is a "right" peak amplitude–it does not matter what the asymmetry is, the condition average peak amplitude at the FuFA peak is always the same, statistically speaking.

## Temporal correlations–simulations

The third of the candidate safety properties suggested by the simulations of [5], is avoidance of temporal correlations between data samples. As previously discussed, in the context of ERP analysis, this issue does not concern correlations along the trial (or ERP) time-series, since the unit of replication is a trial, not a time-point within a trial (see discussion in Supporting

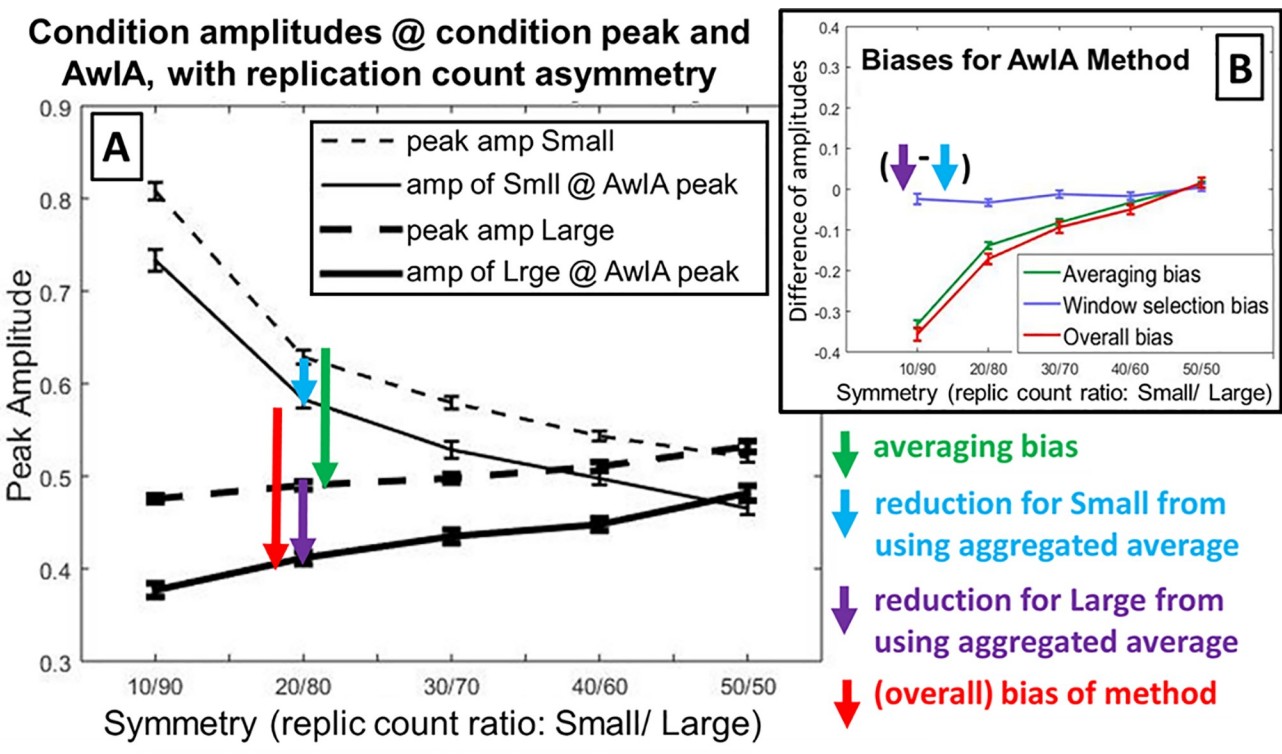

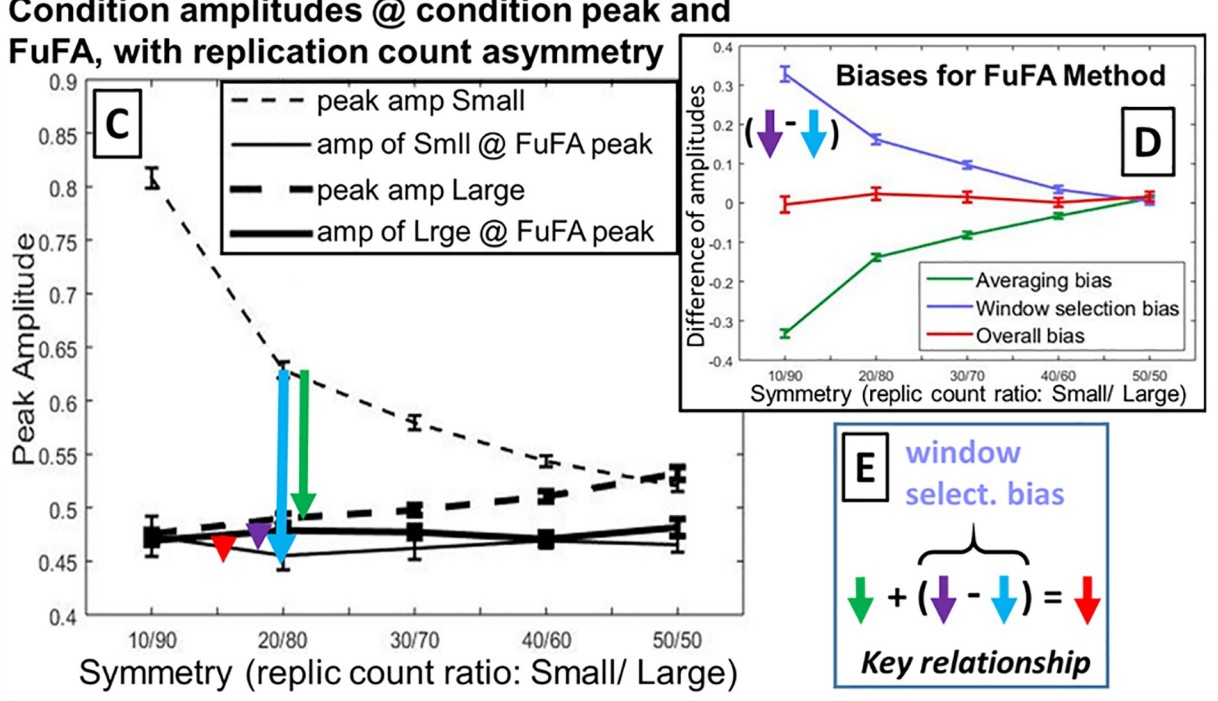

**Fig 6. Results of simulations.** The null-hypothesis was simulated for five replication count asymmetries, from highly unbalanced (10/90) to fully balanced (50/50), with the dependent measure being peak amplitudes of condition averages. **Panels A** and **B** show results for AwIA, while **Panels C** and **D** show results for FuFA. **Panels A** (for AwIA) and **C** (for FuFA) show the main results. *Dashed lines* show peak amplitudes for Small and Large, i.e. when the peak amplitude is read-directly off from the condition averages, without any involvement of an aggregated average. The difference between these lines is the (simple) averaging bias (see green arrow), which is identical for AwIA and FuFA and in both cases, reduces to zero when replication

counts are balanced (50/50). *Solid lines* show amplitudes for Small and Large, when the peak's location is selected from the aggregated average (AwIA for Panel A and FuFA for Panel C). Thus, the difference between solid lines is the overall bias of the method (as indicated by red arrows). In (C), these sit on top of each other, showing that there is no overall bias, while only when replication counts are equalised (i.e. 50/50), do the solid lines coincide in (A). We show, with blue and purple arrows, the amount the amplitude is reduced as a result of going via the aggregated average. Each of these is presented as a reduction,i.e. how much less the amplitude is at the time-point found from the aggregated average than at the true condition peak. The length of the blue and purple arrows reflects the degree to which the aggregated average is "like" Small or "like" Large. As illustrated in Fig 4, the AwIA is more like the Small condition, while the FuFA is more like the Large condition. Accordingly, the reduction due to AwIA (see Panel A) is less for Small than for Large, while the reduction due to FuFA (see Panel C) is dramatically more for Small than for Large. In both cases, this difference in reductions itself reduces until parity is reached at full balance (50/50), see Panels A and C. This *difference* in these two reductions (one for Small, the other for Large) is the *window selection bias*. As previously indicated, the (overall) bias (i.e. difference between solid lines) due to employing an aggregated average process is shown with the red arrows. For the AwIA, Panel A, this (overall) bias is substantial at large replication-count asymmetries, but as would be expected, progresses to zero with fully balanced designs. In contrast, for the FuFA, save for sampling error, there is no (overall) bias at any asymmetries. **Panels B** and **D** summarise biases for AwIA (respectively FuFA). The (simple) averaging bias is the same for AwIA and FuFA, see green arrows and lines. But, while the window selection bias (difference of amplitude reductions, Large minus Small; light purple line), has a small effect in the same direction as the averaging bias for AwIA, it is *equal and opposite* to the averaging bias for FuFA. The overall bias, red arrows and lines, is substantial with large replication-count asymmetries for AwIA, but absent for all replication-count asymmetries for FuFA. Standard errors of the mean are shown. **Panel E**: *Overall bias* is the sum of the *(simple) averaging* bias and *window selection* bias (which itself is a difference of reductions for Large and Small).

Information S1 Text). Thus, with careful experimental design and high-pass filtering of the unsegmented data, in most cases, it should be possible to avoid dependencies from trial-to-trial and thus between data samples, e.g. the mean amplitude in the same window in different trials. However, for completeness, we present simulations here that consider whether temporal correlations are the problem they are suggested to be by the third of Kriegeskorte et al's candidate safety properties.

Clarifying this issue can have value for the cases in which temporal correlations along replication data samples are unavoidable. For example, there can be carry-over effects from trial-to-trial due to learning through the course of an experiment, or perhaps because of the presence of low frequency components, such as the contingent-negative variation, e.g. Chennu et al [18]. In particular, it may be that the presence of such low-frequency components has relevance to the experimental question at hand, rendering it inappropriate to filter them out.

We focus specifically here on a simple case in which correlations are consistent throughout the course of the experiment. To simulate this, we simply *smooth down the replication* data samples at each time-space point of our data segment. That is, for each time-space point, there will be as many replication data samples as there are time-series replications in the experiment, and we convolve these replications with a Gaussian kernel (using matlab command "qausswin" over 6 time points) in a sequence defined by the order in which replication time-series were generated in the simulation. We interpret this as the order replication time-series arose in the experiment.

In more detail, our basic simulation framework is unchanged from that presented in section "**Simulations of FuFA and AwIA**" with the following exceptions.

1. As just discussed, we smooth down replications at each time-space point.

2. We employ a repeating design matrix, which is divided into blocks, such that each block contains 10 replications; see Fig 7.

3. To implement replication-count asymmetry, each block itself is subdivided as follows: 10/90: 1 Small replication, 9 Large replications; 20/80: 2 Small & 8 Large replications; 30/70: 3 Small & 7 Large replications; 40/60: 4 Small & 6 Large replications; and 50/50: 5 Small & 5 Large replications, where in each of these cases, the number of replications for Small equals $N_1$ in Fig 7, and the number for Large $N_2$. In all cases, there are 10 blocks overall.

4. Both aggregated average of peak methods are run, FuFA and AwIA, thereby identifying a (time-space) position of peak for FuFA and for AwIA.

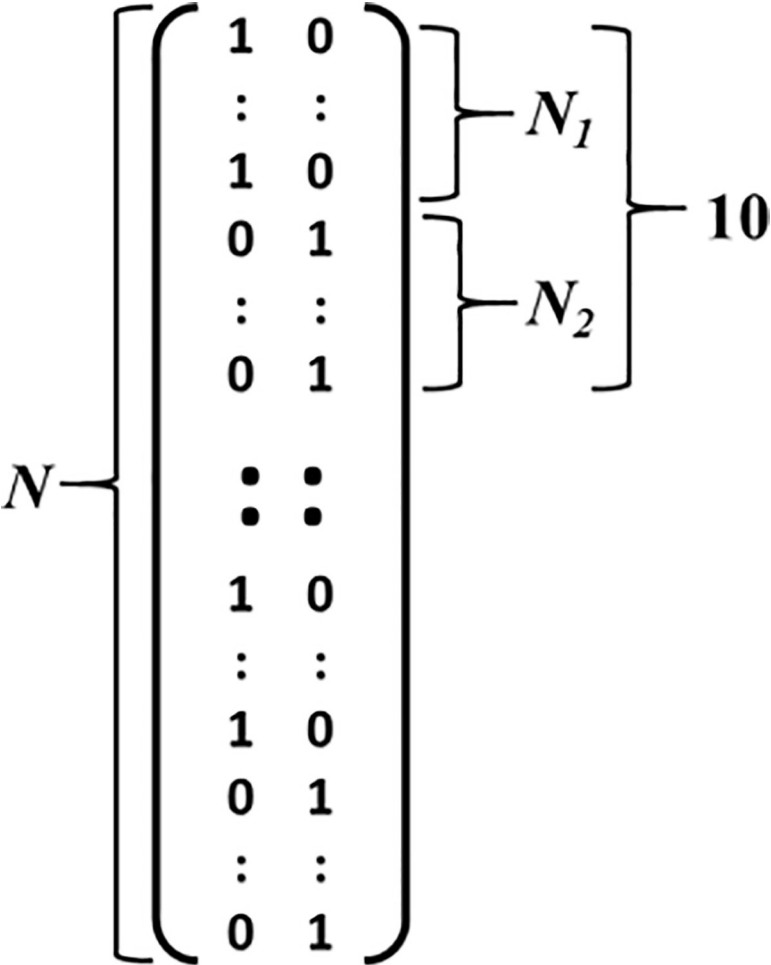

**Fig 7. Form of design matrices used in simulations.** The matrices have a repeating structure, with 10 replications per block. Each block contains $N_1$ replications of condition Small followed by $N_2$ of condition Large. The proportion of $N_1$ to $N_2$ is varied to simulate replication-count asymmetry, from $N_1 = 1$ and $N_2 = 9$ (high asymmetry) to $N_1 = 5$ and $N_2 = 5$ (fully symmetric).

5. Amplitudes are calculated from the Small average and the Large average at the position of the peak of both FuFA and AwIA identified under 4) above.

The results of these simulations are shown in Fig 8. These simulations show very similar patterns to those in Fig 6 –compare panel A in Fig 8 with panel A in Fig 6, and panel B in Fig 8 with panel C in Fig 6. In particular, the overall measure of interest is the difference between the two solid lines (the condition amplitudes at the aggregated average peaks), which show evidence of an asymmetry bias for the AwIA (panel A), but not for the FuFA (panel B). Thus, in the specific smoothing case considered here, we found no evidence that temporal correlations generate a bias beyond that already present with unbalanced designs for the AwIA method. In particular, no evidence of a bias was found for either AwIA or FuFA when replication-counts were balanced (the 50/50 case, furthest to the right on the x-axis in Fig 8), which was the case considered in the simulations by Kriegeskorte et al [5]. We consider this disparity between our findings and Kriegeskorte et al's further when we seek to generalise the simulation results presented here, with a proof of the bias-freeness of the FuFA method with constant temporal correlations in Supporting Information S6 Text.

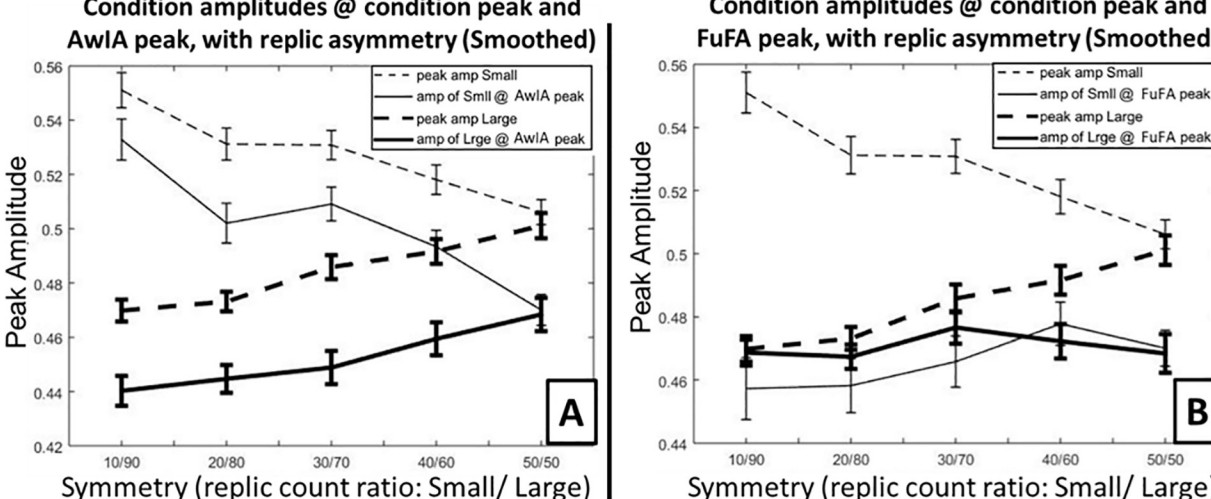

**Fig 8. Results of simulations with smoothing down replication data samples.** The null-hypothesis was simulated for five asymmetries, from highly unbalanced (10/90) to fully balanced (50/50). **Panel A** shows results for AwIA, and **Panel B** results for FuFA. In both panels, *dashed lines* show peak amplitudes for the two conditions, Small and Large, i.e. when the peak amplitude is read-directly off from the condition average, without any involvement of an aggregated average. The difference between these lines is the (simple) averaging bias, which is identical for AwIA and FuFA and in both cases, reduces to zero when replication-counts are balanced (50/50). *Solid lines* show amplitudes for Small and Large, when the location of the amplitude is selected as a peak from the aggregated average (AwIA for Panel A and FuFA for Panel B). In sum, the smoothing employed here has had little effect on the major patterns present in these figures. That is, the AwIA (panel A here) still exhibits a bias, which increases with replication-count asymmetry (i.e. moving from right to left along x-axis), while there is no apparent bias for FuFA (panel B here) at any asymmetry. This can be seen by comparing solid lines (condition amplitudes at aggregated average peaks), the difference of which is the overall bias.

## Why the FuFA is unbiased–formal treatment

We present a mathematical verification that the FuFA approach is bias-free in key situations, and that the AwIA is only bias-free when the design is balanced.

The formal treatment is framed in terms of the general linear model (Eq 1) and its ordinary least squares solution (Eq 2):

$$y = Xb + e \qquad \text{(Eq 1)}$$

$$\hat{b} = (X^T X)^{-1} X^T y \qquad \text{(Eq 2)}$$

where $b$ and $\hat{b}$ are $P{\times}1$ parameter vectors, $X$ an $N{\times}P$ design matrix, $y$ an $N{\times}1$ data vector and $e$ an $N{\times}1$ error vector. Thus, there are $P$ parameters and $N$ data samples. $\hat{b}$ is the inferred estimate of the parameters, $b$.

Then, as per our discussion in section **"Notation"**, $c_s$ is the selection contrast weight vector, which defines the contrast used to select a window, and $c_t$ is the test contrast weight vector.

We focus on the 2-sample independent t-test. Consequently, $c_t$ is the t-test contrast weight vector, i.e.

$$c_t = [+1, -1]$$

Then, for selection contrasts, we introduce the FuFA selection contrast weight vector, which performs a weighted average.

$$c_{s,FuFA} = c_{s,FA} = [N_1/N, N_2/N]$$

where the smaller condition is down weighted, compared to the bigger one, ensuring that each

replication (whether in larger or smaller condition) contributes similarly to the aggregated average. Finally, the AwIA selection contrast weight vector is defined as,

$$c_{s,AwIA} = c_{s,IA} = [1/2, 1/2]$$

In the general case, the application of two contrasts, $c_1$ and $c_2$, will be *parametrically contrast orthogonal* if and only if,

$$c_1 cov(\hat{b}) c_2^T = 0$$

That is, the covariance between parameters, as expressed by the $P{\times}P$ covariance matrix $cov(\hat{b})$, defines the dependencies between inferred parameters, which determine how the application of the two contrasts can impact each other. Note, parametric contrast orthogonality (see Cox & Reid [23] for a discussion of parametric orthogonality) encapsulates the property that even if two parameters covary, if that dependence is irrelevant to the "interplay" between the two contrasts being applied, orthogonality can still obtain.

From here, under ordinary least squares, we can use Eq 2 to derive the following,

$$cov(\hat{b}) = \hat{b}\hat{b}^T = ((X^TX)^{-1}X^Ty)((X^TX)^{-1}X^Ty)^T$$

Then, using $(AB)^T = B^TA^T$ and that transpose is an identity operation over a symmetric matrix, which $(X^TX)^{-1}$ will be, we can derive,

$$cov(\hat{b}) = (X^TX)^{-1}X^Tyy^TX(X^TX)^{-1}$$

In the cases we are considering here, the null hypothesis will hold, since the question for this paper is whether the false positive (i.e. type 1 error) rate is inflated. Consequently, we can assume that the data vector, $y$, has a particular form. That is, focussing on the t-test case, there will be no difference of means between the two conditions, apart from due to sampling error. Accordingly, the term $yy^T$ will generate the data covariance matrix of error noise in the data (which might be generated by pooling errors across space (electrodes) or time-space points). We denote this $N{\times}N$ matrix, where $N$ is the number of replication samples, as $\Sigma$, i.e.

$$yy^T = \Sigma$$

From here, we can give the key orthogonality property, which is as follows.

**Proposition 1.** Under the null hypothesis, *parametric contrast orthogonality* holds between $c_1$ and $c_2$ if and only if $c_1 cov(\hat{b}) c_2^T = 0$, which holds, if and only if,

$$c_1(X^TX)^{-1}X^T\Sigma X(X^TX)^{-1}c_2^T = 0 \tag{Eq 3}$$

As previously discussed, in standard ERP analyses (with EEG or MEG), inference is across replications, not time-points within a trial (or along the entire, unsegmented, time-series of an experiment, as is typical of fMRI analyses). In this context, unless temporal correlations have been elicited between replications through the experiment time-course (e.g. due to learning effects), $\Sigma$ would be a diagonal matrix (i.e. with all off-diagonal elements zero, reflecting the absence of correlations between different replication samples). In this context, parametric contrast orthogonality reduces to the following equation (see the proof of proposition 2 for this derivation).

$$c_1(X^TX)^{-1}c_2^T = 0 \tag{Eq 4}$$

As previously discussed, for completeness, we will also include a consideration of the consequences of temporal correlations across replications; see Supporting Information S6 Text.

## Unbalanced block design matrices

Following on from our simulation results in section **"Simulations of FuFA and AwIA",** we mathematically verify the main results concerning freedom from bias in unbalanced designs, with two "block" design matrices. Thus, we show here that our simulation results generalise, by proving that in all relevant cases, the pattern we observed in our simulations holds. We will do this by showing that Eq 3 holds for $c_{s,FA}$ for all cases we consider, while for $c_{s,IA}$ it only holds with balanced designs.

We assume a design matrix, $X$, of the form,

$$X = \begin{pmatrix} 1 & 0 \\ \vdots & \vdots \\ 1 & 0 \\ 0 & 1 \\ \vdots & \vdots \\ 0 & 1 \end{pmatrix}$$

where the first column is the indicator variable for condition 1 and the second for condition 2. $X$ has $N$ rows, which can be divided into two blocks–upper for condition 1 and lower for condition 2. In the balanced case, these two blocks have the same number of rows: $N/2$, while in the unbalanced case, the upper block has $N_1$ rows and the lower $N_2$, such that $N_1+N_2 = N$. Without loss of generality, we assume that $N_1 \leq N_2$. For example, the following design matrix indicates three replication data samples in condition 1 and four in condition 2.

$$X = \begin{pmatrix} 1 & 0 \\ 1 & 0 \\ 1 & 0 \\ 0 & 1 \\ 0 & 1 \\ 0 & 1 \\ 0 & 1 \end{pmatrix} \quad \text{(example 1)}$$

## Proposition 2

Consider a 2-sample independent t-contrast, with contrast vector $c_t$, in which the noise in the two conditions is generated from the same stochastic process, replications are statistically independent of one another and $X$ is a two block design matrix in which $N_1 \leq N_2$. Then, under the null-hypothesis, parametric contrast orthogonality, i.e. Eq 3, holds for the FuFA, i.e.

$$c_{s,FA}(X^T X)^{-1} X^T \Sigma X (X^T X)^{-1} c_t^T = 0$$

That is, window selection via the FuFA does not bias the statistical test.

**Proof.** Assume a two-block design matrix, such as that shown in example 1. Lack of temporal correlations down replications ensures there is no loss of generality associated with assuming a two-block design matrix.

We first note that Eq 3 can be significantly simplified. Since there are no temporal correlations down replications, $\Sigma$, the data covariance matrix, has a very simple form. Specifically, it is an $N \times N$ diagonal matrix, with the variance of the white noise giving the elements on the

main diagonal.

$$\Sigma = \begin{pmatrix} \sigma^2 & \cdots & 0 \\ \vdots & \ddots & \vdots \\ 0 & \cdots & \sigma^2 \end{pmatrix} = \sigma^2 \begin{pmatrix} 1 & \cdots & 0 \\ \vdots & \ddots & \vdots \\ 0 & \cdots & 1 \end{pmatrix}$$

Eq 3, then, simplifies as follows,

$$c_{S,FA}(X^T X)^{-1} X^T \Sigma X (X^T X)^{-1} c_t^T$$

$$= [\textit{Substitution and scalar multiplication of matrices}]$$

$$\sigma^2 c_{S,FA}(X^T X)^{-1} X^T X (X^T X)^{-1} c_t^T$$

$$= [AA^{-1} = I]$$

$$\sigma^2 c_{S,FA}(X^T X)^{-1} c_t^T$$

We need to show then that $\sigma^2 c_{S,FA}(X^T X)^{-1} c_t^T = 0$, which holds if and only if $c_{S,FA}(X^T X)^{-1} c_t^T = 0$. We do this by simply evaluating the left hand side of this equation.

So, assuming the upper block of $X$ contains $N_1$ rows, the lower block $N_2$ and $N = N_1 + N_2$, we have,

$$(X^T X)^{-1} = \left( \begin{pmatrix} 1 & .. & 1 & 0 & \cdots & 0 \\ 0 & .. & 0 & 1 & \cdots & 1 \end{pmatrix} \begin{pmatrix} 1 & 0 \\ \vdots & \vdots \\ 1 & 0 \\ 0 & 1 \\ \vdots & \vdots \\ 0 & 1 \end{pmatrix} \right)^{-1} = \begin{pmatrix} N_1 & 0 \\ 0 & N_2 \end{pmatrix}^{-1} = \begin{pmatrix} \dfrac{1}{N_1} & 0 \\ 0 & \dfrac{1}{N_2} \end{pmatrix}$$

with which we can derive the result we seek through substitution and evaluation.

$$c_{S,FA}(X^T X)^{-1} c_t^T = \begin{pmatrix} \dfrac{N_1}{N} & \dfrac{N_2}{N} \end{pmatrix} \begin{pmatrix} \dfrac{1}{N_1} & 0 \\ 0 & \dfrac{1}{N_2} \end{pmatrix} \begin{pmatrix} +1 \\ -1 \end{pmatrix} \quad \text{(line XX)}$$

$$= \begin{pmatrix} \dfrac{1}{N} & \dfrac{1}{N} \end{pmatrix} \begin{pmatrix} +1 \\ -1 \end{pmatrix} = \begin{pmatrix} \dfrac{1}{N} - \dfrac{1}{N} \end{pmatrix} = 0$$

(Note also that this derivation can be linked to the idea of two counteracting biases highlighted earlier in this paper; see discussion in Supporting Information S4 Text.)

**QED.** This result demonstrates that Kriegeskorte et al's [5] identification of unbalanced designs as a hindrance to obtaining orthogonality of test and selection contrasts is resolved by employing the FuFA, rather than the AwIA.

We can also show that parametric contrast orthogonality only holds for the AwIA when $N_1 = N_2$.

## Proposition 3

Consider a 2-sample independent t-contrast, with contrast vector $c_t$, in which the noise in the two conditions is generated from the same stochastic process, replications are statistically independent of one another and $X$ is a two block design matrix in which $N_1 \leq N_2$. Then, under the null-hypothesis,

$$c_{s,IA}(X^TX)^{-1}X^T \Sigma X (X^TX)^{-1} c_t^T = 0 \text{ if and only if } N_1 = N_2.$$

i.e. the AwIA approach is only unbiased for balanced designs.

## Proof

This proof follows the deductions of the proof of proposition 2 up to line XX, where we have,

$$c_{S,IA}(X^TX)^{-1} c_t^T = 0$$

From here, we can derive the following,

$$c_{S,IA}(X^TX)^{-1} c_t^T = 0$$

$$\Leftrightarrow [\textit{Derivations in proposition 1 proof and definition of AwIA}]$$

$$\begin{pmatrix} \frac{1}{2} & \frac{1}{2} \end{pmatrix} \begin{pmatrix} \frac{1}{N_1} & 0 \\ 0 & \frac{1}{N_2} \end{pmatrix} \begin{pmatrix} +1 \\ -1 \end{pmatrix} = 0$$

$$\Leftrightarrow [\textit{Manipulations}]$$

$$\begin{pmatrix} \frac{1}{2N_1} - \frac{1}{2N_2} \end{pmatrix} = 0$$

$$\Leftrightarrow [\textit{Manipulation}]$$

$$N_1 = N_2$$

## QED

Finally, do note that although the FuFA approach is parametrically contrast orthogonal, as shown in proposition 2, the contrast weight vectors are not orthogonal, unless the design is balanced, viz, $c_{S,FA} c_t^T = 0 \Leftrightarrow \frac{N_1}{N} - \frac{N_2}{N} = 0 \Leftrightarrow N_1 = N_2$. Accordingly, the first proposed safety property of Kriegeskorte et al [5] is not strictly required.

## Statistical power

A central concern of this paper is the type-I error rate. With classical frequentist statistics, maintaining the false positive rate of a statistical method at the alpha level ensures the soundness of the method. A failure to control the type-I error rate is what is suggested by a replication crisis, i.e. results are being published with the stamp of significance against a standard 0.05 threshold, however, the percentage of published studies that do not replicate is much larger than 5%.

Statistical power (one minus the type II error rate) is, of course, also important; that is, we would like a sensitive statistical procedure that does identify significant results, when effects are present. This is the question that we consider in this section. Specifically, we extend the assessment of statistical power made in Brooks et al [8]. In these new simulations, there is no trial-count asymmetry, as a result, in this section, we talk in terms of the aggregated average, rather than the FuFA, since FuFA and AwIA are the same in this context.

Brooks et al [8] provided a simulation indicating that the aggregated average approach to window selection is more sensitive than a fixed-window prior precedent approach when there is latency variation of the relevant component across experiments. This is, in fact, an obvious finding: with a (fixed-window) prior precedent approach, the analysis window cannot adjust to the presentation of a component in the data, but it can for the aggregated average/ FuFA.

A more challenging test of the aggregated average's statistical power is against mass-univariate approaches, such as, the parametric approach based on random field theory implemented in the SPM toolbox [15] or the permutation-based non-parametric approach implemented in the Fieldtrip toolbox [16,17]. This is because such approaches do adjust the region in the analysis volume that is identified as signal, according to where it happens to be present in a data set. However, because mass-univariate analyses familywise error correct for the entire analysis volume, their capacity to identify a particular region as significant reduces as the volume becomes larger. In contrast, the aggregated average approach is not sensitive to volume size in this way, implying that it could provide increased statistical power, particularly when the volume is large.

This is the issue that we consider in simulation in this section. Specifically, we take this paper's main data generation approach, map it to the 10–20 electrode montage that is standard in EEG work, and then compare the statistical power of Fieldtrip's cluster inference procedure and the aggregated average approach. The decision to focus on a cluster-based permutation test reflects the method's prominence in EEG/MEG research, where it is effectively a de facto standard.

Details of the simulations are as follows.

We generated simulated EEG data, in the way described earlier (c.f. subsection "**Construction of Simulations**" in section "**Unbalanced Designs–Simulations**" and subsection "**Simulations of FuFA and AwIA**") with the following changes.

1. A 9x9, rather than 8x8, spatial grid is used, since it is more naturally mapped to the 10–20 system, with the centre of the grid mapped to Cz.

2. Signal time-series were included in the centre of the grid, at positions 4,4; 4,5; 4,6; 5,4; 5,5; 5,6; 6,4; 6,5; and 6,6.

3. As previously, we had two conditions; here, each comprised 20 replications. The difference between conditions was generated by scaling the signal in the first condition by 0.2 and the second by 0.15. This contrasts with our other simulations in this paper, in which there was, in a statistical sense, no difference between the two conditions, as the null was being simulated.

4. We spatially smoothed the data with a Gaussian kernel of width 0.8; this meant that taking the peak in our aggregated average approach reflected an integration over a relatively broad region of the scalp.

5. We mapped the 9x9 spatial grid to the 10–20 electrode montage as follows,

   a. Grid position 4,3 to Fp1; 5,3 to Fpz; 6,3 to Fp2; 3,4 to F7; 4,4 to F3; 5,4 to Fz; 6,4 to F4; 7,4 to F8; 3,5 to T7; 4,5 to C3; 5,5 to Cz; 6,5 to C4; 7,5 to T8; 3,6 to P7; 4,6 to P3; 5,6 to Pz; 6,6 to P4; 7,6 to P8; 4,7 to O1; 5,7 to Oz; and 6,7 to O2.

Grid locations not mapped to an electrode were discarded.

Examples of the time-domain data generated by our simulations are shown in Fig 9.

We then performed the following analyses on each simulated data set.

1. We first performed a time-domain analysis on the simulated data, in the fashion discussed in section **"Simulations of FuFA and AwIA".**

2. We then performed a time-frequency decomposition of the simulated data in Fieldtrip. As an illustration, in Fig 10, we show the results of our frequency domain analysis of the data presented in Fig 9.

3. The time-frequency analysis had the following properties.

    a. We filtered to identify the 3 to 30 hz frequency range.

    b. Wavelet decomposition was performed, with a five cycle wavelet.

    c. To enable low-frequency wavelet estimation, we pre-pended and post-pended buffer periods of coloured noise according to the human frequency spectrum; see Supporting Information S7 Text. For both pre- and post-pending, these periods were twice the length of the main analysis segment.

    d. We used the Fieldtrip "absolute" baseline correction, which was applied in the 100ms time period before stimulus onset.

4. We performed the same statistical inference procedure on both time and frequency domains.

5. At the first (samples) level, we performed a two-sample independent t-test and then, at the second level, we applied a cluster-based familywise error correction, with Monte-Carlo resampling (2000 resamplings), according to the Fieldtrip electrode neighbourhood template eiec1020_neighb.mat.

6. For the cluster inference, the result of each simulated data set that we were interested in was the p-value of the largest positive cluster mass.

7. The aggregated average was constructed by taking the union of replications of the two conditions and then averaging (note, there was no trial-count asymmetry, so this is the same as averaging the average of each condition, hence the FuFA and AwIA's are not different here). The time-space point of the maximum amplitude in this average was taken as the ROI in the time-domain. The same basic procedure was performed in the frequency domain, although only after a time-frequency analysis was performed on the union of replications. In this case, the selected ROI was the time-space-frequency position of the maximum power in the resulting volume.

8. The aggregated average result of each simulated data set was the uncorrected p-value of the two-sample independent t-test at the selected point/ROI on the aggregated average.

The results we report are from 40 runs of the simulation code and, as a result, we show 40 data points for each of the simulation conditions we explore. These conditions were time domain+-aggregated; time domain+cluster; frequency domain+aggregated; and frequency domain+cluster.

Our results are presented as probit-transformed p-values. Probit maps p-values to a minus to plus infinity range, enabling differences between small p-values to be easily observed. Results are shown in Fig 11. Panel A provides the main summary of our findings. We can see that the two aggregated conditions exhibit more extreme negative going probit values, and the difference between aggregated and cluster was larger in the frequency domain.

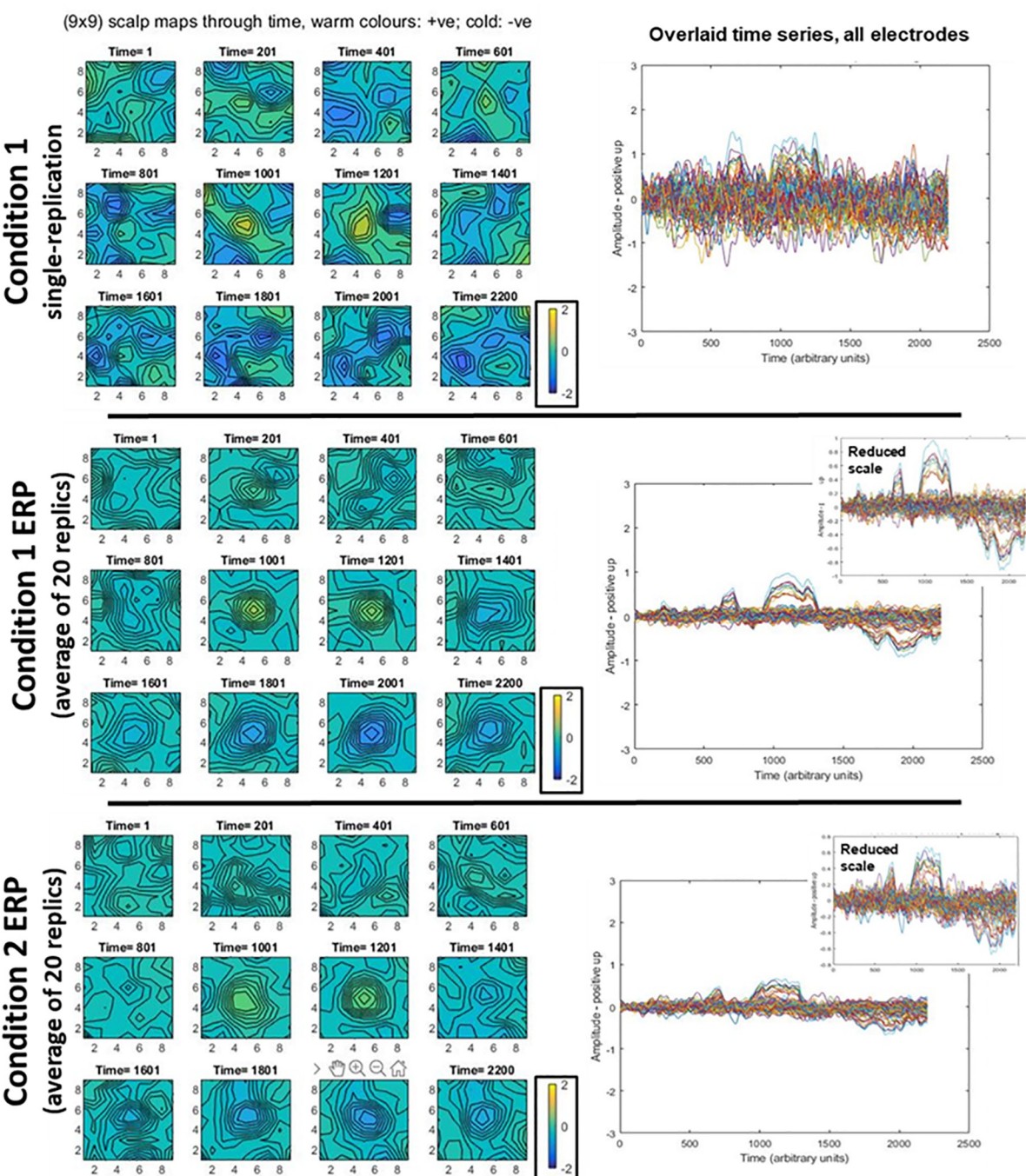

**Fig 9. Illustrative data generated for statistical power simulations.** In all rows, we present the same EEG data in two different ways. On the left, scalp topographies through time are presented, with all three topography sequences using the same colour scale to aid comparison. On the right, time-series at each electrode are presented overlaid in the same plot. The first row shows a typical singe-replication for condition 1; the same plot for a condition 2 replication would look similar, since the amplitude difference of the signal is swamped by noise. The second row shows a typical condition 1 average (ERP), here generated from 20 replications and the third row shows the same, but for condition 2. All the main time-series plots have the same scales to aid comparison between amplitudes of a single replication and averages. As would be expected, the single replication contains much more extreme deflections (both positively and negatively). This can be seen in the more extreme colours in the top-row scalp topographies, and the larger amplitudes in the corresponding overlaid time-series plot. The reduction in extreme amplitudes evident on the right side due to averaging, has enabled the signal to emerge. This can be seen as a positive deflection at the centre of the grid, at time-points 1001 and 1201, and a negative one also at the centre of the grid, in the time-range 1801–2200. As would be expected, the overlaid time-series plot of the average shows the signal landmarks in the same time periods, see particularly, inset plots on the right. Condition 1 has higher signal amplitude than condition 2.

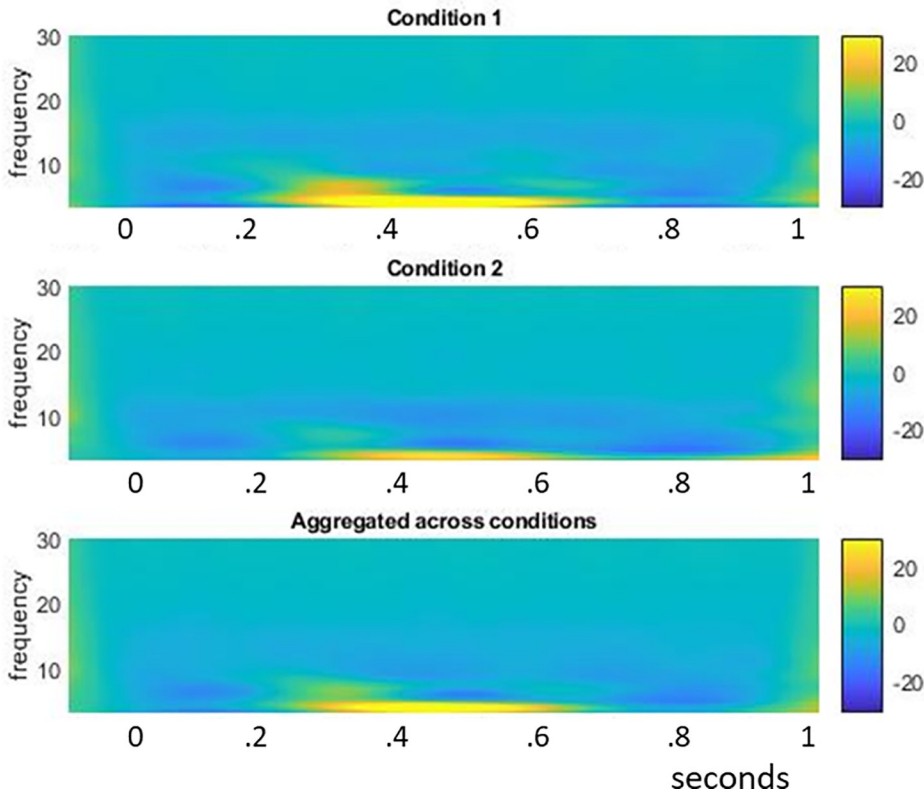

**Fig 10. time-frequency plots of example statistical power data simulations.** We show typical plots of condition 1 and condition 2, as well as of the aggregated average. As can be seen, since the main time-frequency feature appears at the same point for both condition 1 and condition 2, the aggregated average plot also reflects this dominant feature.

We also run a 2x2 ANOVA with probit-transformed p-values as dependent variable, and factors domain (time vs frequency) and method (aggregated vs cluster). The main effect of domain was not significant ($F(1,156) = 0.44$, $p = 0.51$, partial_eta$^2$ = 0.0027), but the main effect of method was highly significant ($F(1,156) = 57.51$, $p<0.0001$, partial_eta$^2$ = 0.2610), and the 2x2 interaction also came out significant ($F(1,156) = 5.9$, $p = 0.0163$, partial_eta$^2$ = 0.0349). These findings are consistent with the box-plots. In particular, the effect sizes (which are not dependent upon the number of simulated data sets generated, which is effectively arbitrary and could be easily extended) showed a large effect of method, with the aggregated average exhibiting substantially more statistical power (i.e. lower p-values for the same data set), and also an interaction that suggests that the benefit of the aggregated average approach is larger for the frequency than the time domain.

The findings here serve as a proof of principle that the aggregated average approach can increase statistical power over cluster-based FWE-correction, which is the de facto standard in the field. In addition, and perhaps most importantly, the aggregated average approach maintains its statistical power when an extra dimension (here frequency) is added to the analysis volume. This is not a surprising finding, since the statistical power of cluster-inference falls as the analysis volume increases in size. This is simply because the probability of a particular size of (observed) cluster arising under the null increases as the volume increases.

On the other hand, the aggregated average approach presented here will not do well if an effect exhibits a polarity reversal between conditions. Indeed, cluster-inference could find a large effect when for a particular period, condition 2 is -1 times condition 1. In contrast, the

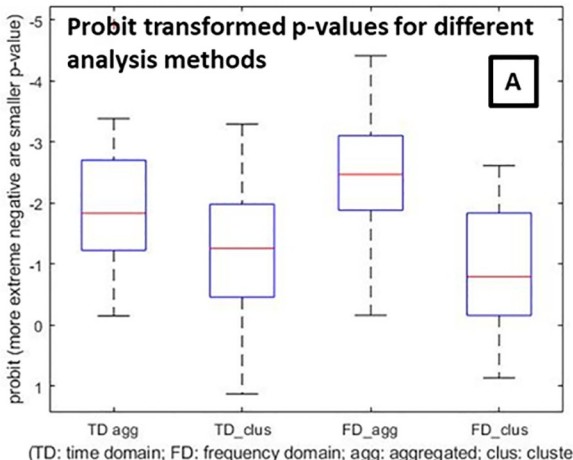

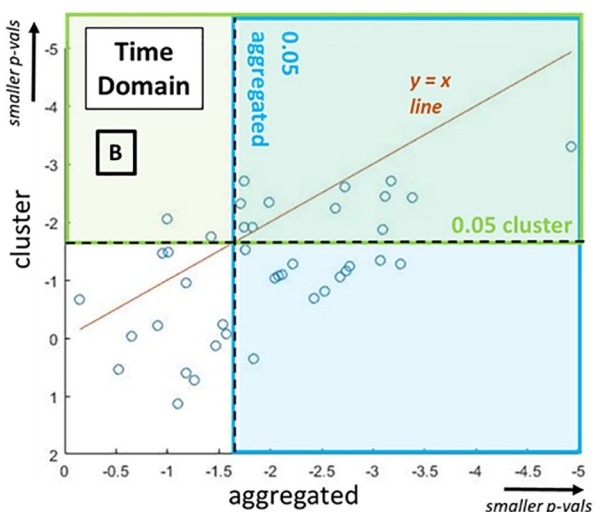

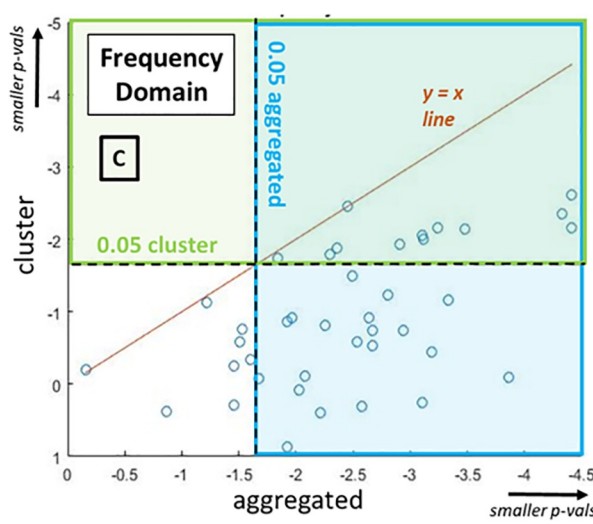

**Fig 11. Simulation results, expressed as probit transformed p-values.** [A] Main results depicted as box-plots for time-domain aggregated average, time-domain cluster-based analysis, frequency domain aggregated average and frequency domain cluster-based analysis. Red markers indicate the median; bottom and top edges of boxes indicate the 25th and 75th percentiles, respectively; whiskers extend to most extreme non-outlier data points; and "+" symbols mark outliers. [B, C] Scatter plots show that, as one would expect, the aggregated average and cluster analysis generate correlated results. Note, the brown line is not a line of best fit, it is simply the identity line: Y = X. Data sets in which the aggregated average gives a smaller p-value than cluster inference appear below the Y = X line and those where cluster inference does better appear above it. The 0.05 p-value threshold corresponds to a probit transformed value of -1.6449. We show where this threshold sits with green and blue dashed lines. As a result, the points in the green region are significant by cluster inference and blue by aggregated average. Time domain aggregated has 25/40 significant, time domain cluster has 14/40, frequency domain aggregated has 32/40, and frequency domain cluster has 12/40. These scatter plots show again that, for these simulations, the aggregated average is more effective, giving more statistical power, than cluster-inference, and that this is especially the case in the frequency domain.

aggregated average would be zero in that period. Further discussions of the pros and cons, assumptions underlying and usage guidelines for the aggregated average, can be found in Table 4 of Brooks et al [8].

## Discussion

This paper has presented simulated and formal grounding for a simple method, the Fully Flattened Average (FuFA) approach, to place analysis windows in M/EEG data without inflating the false-positive rate. The reason why we believe that the FuFA approach is so effective is

because, as demonstrated, it does not inflate the false positive rate under the null hypothesis, but nonetheless it tends to "pick-out" the ERP components of interest, which often arise at a similar time region in all conditions in a particular experiment. Indeed, the FuFA method works particularly well if the component of interest is strong in all conditions, just with an amplitude (but little latency) difference; see [8] for a demonstration of this. In this way, it keeps the type 2 error rate relatively low. This is confirmed by our statistical power simulations, which showed that with realistic generated EEG data sets, the aggregated average/ FuFA approach has higher statistical power than Fieldtrip cluster-inference. Furthermore, this benefit was even greater when analysis was in the frequency domain, which adds a dimension and thus size to the analysis volume. The results of these simulations reflect the trade-offs with respect to statistical power between the aggregated average and cluster-inference methods. It is, though, certainly the case that the aggregated average will tend to do better when 1) the volume is large, and 2) effects ride on the top of large components, which have the same polarity and similar latencies in different conditions.

For the generality of the results presented, we have considered a broad framing of aggregated averages, thereby enabling our findings to apply whatever the unit of inference–trial, participant, item, etc. Our previous article on the problem of window and ROI selection [8], though, specifically focussed on inference across participants and placing windows on the grand average across all participants. To make the link to this earlier work completely clear, if participants are the unit of inference, the FuFA becomes the Aggregated Grand Average of Trials and the AwIA becomes the Aggregated Grand Average of Grand Averages, the concepts discussed in [8].

With regard to the generality of the FuFA approach, it is important to note that it applies as much to within as it does to across participant designs. Our work concerns the number of trials/ repetitions that are incorporated into an average, i.e. in an Event Related Potential (ERP). Even though statistics are run at the participant level, the ERP for each participant is generated by averaging trials. If there are disparities of trail-counts entering these averages, the problem we highlight will still obtain with a within-participant design. To put it in other terms, although statistical inference is performed on participant-level observations, observations at that level are generated from observations at the trial-level, where asymmetries of observation counts can arise.

As an illustration, imagine a simple within-participants experiment, where we have N participants and two conditions; and all participants complete both conditions. We then run a *paired* t-test, i.e. the simplest within participants test, but we vary the trial-counts going into the ERPs between the two conditions. We obtain the bias shown in Fig 12. Trial count asymmetry runs on the x-axis and false positive rate on the y-axis. As you can see, it does not matter whether the experiment is paired or unpaired, there is always an increasing bias (i.e. increasing false-positive rate) as the asymmetry increases for the averaging that is not flattened (i.e. the AwIA). This bias is eradicated when the flattened average is taken (which is the FuFA approach). The pattern is almost identical for paired and unpaired t-tests, i.e. within or across-participant experiments.

Another way of thinking about the issue is that the amplitude of the noise relative to the signal in a participant-level ERP is affected by the number of trials contributing to that ERP. In this way, trial-level observations impact participant-level observations.

Parametric contrast orthogonality, see Eq 3, gives assurance that selection and test contrasts when applied within the context of a particular general linear model inference are orthogonal. However, in a Human Brain Mapping poster, Ridgway [24] identified an additional pitfall that arises when statistical tests are applied to both the selection and the test contrasts, and which corresponds to a difficulty previously identified in the statistics literature [25]. The essence of the problem is that even if the inferred selection and test contrasts are parametrically orthogonal, non-orthogonality can creep back in through the error variance. For example, if windows/

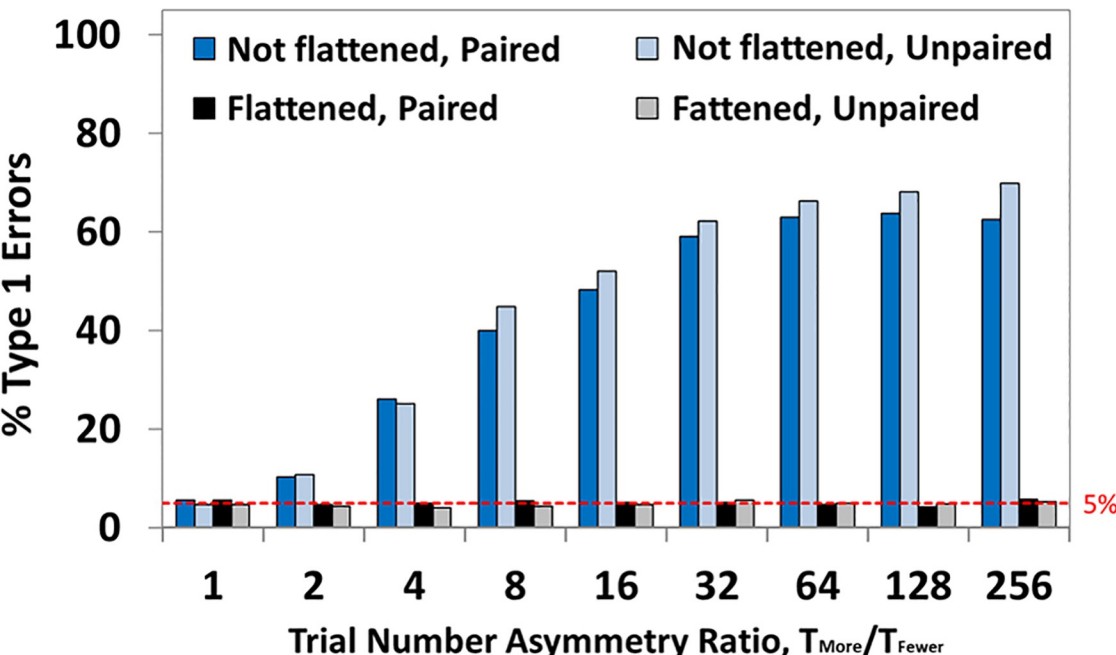

**Fig 12. Results of simulation of null, incorporating a within-participant test.** The simulation involved two levels of noise. The inter-trial noise source was independently generated on each trial, but the same algorithm was used across trials, participants, and conditions (see Brooks et a [8]). Inter-participant noise was generated independently for each participant. The exact same noise was added to every trial (in both conditions) for the participant. The results of this simulation (noise-only data) clearly showed that the pattern of Type I error rates was not substantially different between paired and unpaired data sets (compare dark bars to lighter coloured bars). There is clear evidence of inflation of the false positive rate when a non-flattened average is taken (i.e. the AwIA). This inflation is eradicated when the flattened average (i.e. the FuFA) is taken. The plot in this figure is for noise-only data, but we include a similar simulation incorporating a within-participant experiment with a strong N170 signal present in Supporting Information S8 Text. The N170 results again show similar results for paired and unpaired data.

ROIs are selected according to an F-test, and then an F-test is also applied on the test contrast, the denominators of these two F-tests (i.e. the mean squared error) will be driven by the same variance. This biases towards windows/ROIs in which variance is lower, which could arise under the null simply from sampling error. This will reduce test statistic p-values, increasing the rate of false-positives.

This difficulty can, though, be avoided if the error variance does not contribute to the selection of windows/ROIs. For example, selection could be made using an unstandardized effect, e.g. the numerator of an F-test, or the application of a simple contrast, which is the approach focussed on in this paper.

A further point of note is that the mathematical findings in this paper are more general than the simulation results. Our simulations are specific to selecting extreme values, e.g. the maximum or minimum. That is, our simulation results suggest that the FuFA approach is unbiased specifically in the context of selecting maxima (e.g. peaks) or minima (e.g. troughs). However, the propositions we prove in our formal treatment are statements of the orthogonality of the FuFA and a t-contrast. Thus, it does not matter what landmark one seeks to pick in the FuFA, for example, window selection could focus on zero crossing points, the orthogonality result will still apply.

The most common type of EEG experiment is one in which participants are the random effect. As just discussed, when this is the case, the FuFA becomes an Aggregated Grand Average of Trials, as introduced in [8]. In this context, the typical approach would be to perform window selection at the grand average level. However, in contrast, a different aggregated

average could be determined for each participant, tuning to the data of each participant separately, without requiring a distinct functional localizer [14] or functional profiler [26]. Such an approach is sound, and could, for example, maintain statistical power in the context of high variability in component latency across individuals, but relative consistency within individuals, i.e. across conditions.

Returning to pre-registration, as previously discussed, the registration of fixed windows runs the risk of inflating type II error rates. One obvious solution to this is to allow pre-registration of an orthogonal contrast procedure, with the bounding search region for a particular component pre-specified, but not the actual integration window position. In this way, the benefits of pre-registration with regard to controlling false-positive rates could be combined with a data-driven procedure for window identification to ensure that type II error rates are not dramatically inflated.

We can also think in broader terms about the FuFA procedure and orthogonality in general. Windows/ROIs are just one example of a set of hyper-parameters that need to be set when performing an M/EEG analysis. Other such hyper-parameters include, filter settings; artefact rejection procedures; re-referencing, e.g. to mastoid or ensemble average; frequency bands for a time-frequency analysis; even classifier hyper-parameters, such as type of kernel used (see [9] for a discussion of this). If any such hyper-parameter is optimized to give a desired effect, the false positive rate will be inflated. In essence, the problem is putting the analysis pipeline in a loop with the output of that pipeline, viz p-values, F-values or t-values. Would it be possible, then, to apply the same aggregated average, or more generally, parameterized contrast orthogonalization, to setting these other hyper-parameters? This is an important line for future research.

An alternative way to resolve the problem of post-hoc fishing in analysis hyper-parameters is to partition the collected data, tune hyper-parameters on one part and test on a separate part. In the context considered in this paper, this would amount to selecting windows/ROIs on one part of the data, but then testing and reporting on the other part. And to be clear, with such partitioning, one really can tailor hyper-parameters on one part, without invoking an orthogonal contrast of any kind. This is because, in a statistical sense, the noise in the selection partition is different to the noise in the testing partition, so any advantage obtained by fitting hyper-parameters in one partition to the noise, i.e. over-fitting, will not benefit the testing in the other partition. Classic examples of such data partitioning are functional localisers [27] and cross validation [5].

Certainly, a technique such as cross validation is an important tool in the analysis toolbox, particularly, when there are no precedents at all for the landmarks that should be expected in a data set. In particular, the orthogonality approach breaks down if it is unclear how to even pre-specify the properties of the selection contrast (e.g. the polarity of the component being searched for, or in what general {bounding} region of the analysis segment it might appear in), which for the method to not inflate false-positives need to be pre-defined. However, all data partitioning carries a cost, which is a loss of statistical power. That is, if data sets are split, the final test result to be reported has to be assessed on a subset of the whole data, reducing power. A key benefit of the parametric contrast orthogonality approach is that all data contributes to the reported statistical test. This benefit becomes all the more pronounced as the expense of collecting data increases, e.g. when moving from behavioural experiments to EEG (which is somewhat more expensive) to MEG/fMRI/PET (which are a lot more expensive).

## Conclusions

In the absence of any further explanation, statements in M/EEG papers of the kind, "window was placed according to visual inspection of grand average" should be a "red flag" for reviewers

and readers. At the least, some sort of justification on the basis of prior literature should be given for window/ROI placements.

The FuFA approach, and parametric contrast orthogonalization in general, offers an alternative that enables windows/ROIs to be tuned, in a data-driven manner, to the landmarks of a particular data set without incurring a false positive inflation. The aggregated average approach can be sensitive to replication and noise asymmetries between conditions, but, as verified in this paper, the former is resolved by using the FuFA. In conclusion, then, the FuFA approach provides a method to *dip twice into the data*, *without double dipping in contrast space*.

## Supporting information

**S1 Text. First-level analyses in EEG and fMRI**
(DOCX)

**S2 Text. Neuroimage Clinical, Orthogonal Contrasts**
(DOCX)

**S3 Text. Link to Brooks et al findings**
(DOCX)

**S4 Text. Formal Manifestation of Two-biases**
(DOCX)

**S5 Text. Prior Precedent in ROI Placement–an Example**
(DOCX)

**S6 Text. Repeating Design Matrices and Temporal Correlations**
(DOCX)

**S7 Text. Noise Generation Process**
(DOCX)

**S8 Text. Further Type I Error Simulation Incorporating Within-Participant Design.**
(DOCX)

## Acknowledgments

We would like to thank Karl Friston and Guillaume Flandin for very valuable discussions concerning orthogonal contrasts and their mathematical formulation. We would also like to thank the valuable suggestions from two referees, which have improved the readability and contribution of this paper.

## Author Contributions

**Conceptualization:** Howard Bowman, Joseph L. Brooks, Alexia Zoumpoulaki, Vladimir Litvak.

**Formal analysis:** Howard Bowman, Vladimir Litvak.

**Resources:** Omid Hajilou.

**Software:** Howard Bowman.

**Supervision:** Howard Bowman.

**Validation:** Joseph L. Brooks, Omid Hajilou, Alexia Zoumpoulaki, Vladimir Litvak.

**Visualization:** Howard Bowman.

**Writing – original draft:** Howard Bowman.

**Writing – review & editing:** Joseph L. Brooks, Alexia Zoumpoulaki, Vladimir Litvak.

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
