## [Decision Letter · Decision Letter 0]

10 Oct 2019

Dear Dr Bowman, 

Thank you very much for submitting your manuscript 'Breaking the Circularity in Circular Analyses: Simulations and Formal Treatment of the Flattened Average Approach' for review by PLOS Computational Biology. Your manuscript has been fully evaluated by the PLOS Computational Biology editorial team and in this case also by independent peer reviewers. The reviewers appreciated the attention to an important topic, but they raised substantial concerns about the paper. Based on the reviews and editorial discussions, we regret that we will not be able to accept this manuscript for publication in the journal. 

The reviews are copied below, and we hope they may help you should you decide to revise the manuscript for submission elsewhere. We are sorry that we cannot be more positive on this occasion, but hope that you appreciate the reasons for this decision and that you will consider PLOS Computational Biology for other submissions in the future. 

Thank you again for your support of PLOS Computational Biology and open-access publishing. Please do not hesitate to get in touch (via ploscompbiol@plos.org) if we can provide any further assistance. 

Sincerely, 

Leyla Isik 

Associate Editor 

PLOS Computational Biology 

Samuel Gershman

Deputy Editor

PLOS Computational Biology 

[LINK]

Reviewer's Responses to Questions

**Comments to the Authors: **

Reviewer #1: Dear authors,

Thank you for the opportunity to review your paper.

You discuss the properties of a method for ROI selection. Specifically, you consider the situation in which a researcher that is interested in the difference between two conditions, selects his/her ROI as the maximum of the AVERAGE over these two conditions. You focus on the different properties of ROI selection based on the unweighted and the weighted average (which you denote as AwIA and FuFA) and demonstrate that only ROI selection based on the weighted average (FuFA) is unbiased.

I was not surprised to learn about your result, because I could not see how bias could emerge when using the weighted average, whereas this was quite easy to see for the unweighted average. Your simulations and formal proof now support this intuition.

Your result depends on the fact that there is an unequal number of observations in the different experimental conditions. An unequal number of observations can only happen in a between-participant experiment, because in a with-participants experiment all participants are observed in all conditions. Because most neuroscience experiments are within-participant experiments, your result is only relevant for a minority of the neuroscience experiments. 

I find the paper quite long for the points that you make. It must be possible to bring across the same message by first explaining why the weighted average approach is unbiased, followed by a demonstration of the bias in the unweighted average approach. The unbiased nature of the weighted average approach follows from the fact that the weighted average is unaffected by noise differences between the experimental conditions. Therefore, also the maximum of this weighted average is unaffected by these noise differences.

Reviewer #2: The mathematical treatment and simulations provided make a nice case for double dipping without inflating type 1 error rate. Overall, I think it is an interesting paper providing a good solution to a tangible problem. I have minor concerns described below.

Introduction

- it seems like reproducible, replicable and reliable are used interchangeably - these are not the same ; i suggest from the start pointing out to definitions so the readers knows what you are talking about precisely (eg https://arxiv.org/abs/1802.03311)

- lines 56/57 'For NI studies this would include specifying the ROI' -- this could, depends on hypotheses

- figure 1 could do with topographic plots, as if can also show difference of location and not just latency

- lines 92/93 i would temper the sentence saying that pre-registration makes it difficult to detect novel effects -- ROI pre-registration will indeed inflate type 2 error, but pre-registration doesn't prevent to also do the full brain analysis as exploratory (it simply makes clear the distinction)

- lines 143/144/145 - could not follow that sentence; please rephrase

- lines 192/194 in a two samples case should we have three columns with the constant (say last column like SPM) and thus c = [1 -1 0] and X with a columns of 1

- line 205 use X and not Z to keep with notation

- line 200 ref Pernet et al 2011 (eeg) seems more appropriate than Penny et al 2011 -- or refer to a specific chapter dealing with EEG that way (ie without factoring time)

- lines 230/231/232 why not adding the examples for the reader (dot([1 -1 0],[1/2 1/2 0])= 0 and

dot([1 -1 0],[3/7 4/7 0])= -0.1429)

Simulations:

- lines 297+ lease describe the signal and noise parameter [from the noise.m function I assume noise (frames, epochs, sampling rate))

- line 426 over how many cell of the grid the smoothing was applied? (ie size of the kernel)

- line 586 typo, AwIA valid only when balanced

Discussion:

- line 733++ this statement is only true if you compare apples and oranges such as latencies and/or locations are completely different otherwise latency differences are reflected in the amplitude differences thus your approach is valid in most cases

- SPM can return an orthogonal contrast from another one? which function? it's not in the GUI, doesn't look like spm_FcUtil or spm_SpUtil can return this

- line 786 ; one of the key aspect of pre-registration is determining the number of subjects, and that's why this is typically done on a ROI - or do you suggest if we have data, N could be based on where a 1st experiment saw the effect (biased in location possibly) but window based on FuFA?

- temporal correlation discussion: I think in fmri to assume exact same temporal correlation between trials is harder unless the stimulus presentation order is identical between conditions; mostly due to the autocorrelation of the signal, since the regression is performed in time, unlike erp. Thus in general, I'd think the issue is valid for most event related designs in fMRI.

Dr Cyril Pernet

**Have all data underlying the figures and results presented in the manuscript been provided?**

Reviewer #1: Yes

Reviewer #2: No: the software is there but no access to simulation code or the data generated by the simulations

PLOS authors have the option to publish the peer review history of their article (what does this mean?). If published, this will include your full peer review and any attached files.

Reviewer #1: No

Reviewer #2: Yes: Dr Cyril Pernet

---

## [Decision Letter · Decision Letter 1]

15 May 2020

Dear Prof. Bowman,

Thank you very much for submitting your manuscript "Breaking the Circularity in Circular Analyses: Simulations and Formal Treatment of the Flattened Average Approach" for consideration at PLOS Computational Biology.

As with all papers reviewed by the journal, your manuscript was reviewed by members of the editorial board and by several independent reviewers. In light of the reviews (below this email), we would like to invite the resubmission of a significantly-revised version that takes into account the reviewers' comments. It will in particular be important to address Reviewer 1's remaining concerns about the potential sensitivity enhancement of your method and comparison to cluster-based permutation testing.

We cannot make any decision about publication until we have seen the revised manuscript and your response to the reviewers' comments. Your revised manuscript is also likely to be sent to reviewers for further evaluation.

Sincerely,

Leyla Isik

Associate Editor

PLOS Computational Biology

Samuel Gershman

Deputy Editor

PLOS Computational Biology

Reviewer's Responses to Questions

**Comments to the Authors:**

Reviewer #1: Dear authors,

Thank you for your extensive reply to all the reviewers' comments.

You replied to my comment on the applicability of your main point to within-participant experiments, and I agree with your main point. You also made changes in the main text w.r.t. the possible nature of the replications (participants or trials). However, I stumbled over the following sentences in your reply:

"Specifically, the weighted average one would take in our context would involve weighting condition 1 by the scalar 1/(1 + 2) and condition 2 by the scalar 2/(1 + 2), where is the number of trials in condition i. This would generate the FuFA, in our terminology. Unfortunately, the FuFA is not equally affected by the two conditions, i.e. it is not unbiased."

With these weights applied to the condition-specific averages, you would indeed obtain the FuFA, but this average would result in an UNbiased ROI selection (instead of "not unbiased"), and this is the main point of your paper. I will assume that this is a typo.

As with the original version of your paper, I am not surprised by the fact that an unequal number of trials per condition will create a bias if ROI selection is based on the unweighted average (AwIA). It is another example of the general rule that ROI selection is biased if it is affected differentially by the noise in the two conditions. It is easy to extend this rule further. For example, you would also get this bias in a within-participants study with an EQUAL number of trials per condition, but with trials of an UNequal length. For some reason, you may have time windows [200,250] ms in one condition and [150,300] in another, and prior to performing a statistical analysis, you average over this time window. You may argue that this would be a very unusual procedure, but this also holds for ROI selection based on an unweighted average of condition means that are based on an unequal number of trials.

In my experience as a reviewer, my colleagues are typically very well aware of the possible biases that may result as a consequence of an unequal number of trials in the different conditions. (These biases usually do not pertain to ROI selection, but to bias-sensitive measure like R-square and coherence.) They typically deal with this by asking for control analyses with an equal number of trials in the different conditions.

On the whole, I think that your paper contains valid points. However, it does not focus on the most important point (which should be sensitivity enhancement instead of Type 1 error rate control) and is not written for the most appropriate audience (which should be applied cognitive/medical neuroscientists instead of the more theoretically oriented readers of PLOSCompBiol). I think you could make an important contribution by quantifying the sensitivity enhancement that follows from data-driven ROI selection. To make an impact, it is important to compare your ROI-based approach to the current standard in the field, cluster-based permutation testing (Eric Maris and his colleagues), whose sensitivity decreases with every additional data dimension (space, frequency, time). As a part of a plea for data-driven ROIs, you should point out that bias may occur in case the unweighted average is used for ROI selection (the main point of your current paper).

Reviewer #2: thx for the revision - all my comments were addressed and I'm agree with the changes.

**Have all data underlying the figures and results presented in the manuscript been provided?**

Reviewer #1: Yes

Reviewer #2: Yes

PLOS authors have the option to publish the peer review history of their article (what does this mean?). If published, this will include your full peer review and any attached files.

Reviewer #1: No

Reviewer #2: Yes: Dr Cyril Pernet
---

## [Decision Letter · Decision Letter 2]

24 Aug 2020

Dear Prof. Bowman,

We are pleased to inform you that your manuscript 'Breaking the Circularity in Circular Analyses: Simulations and Formal Treatment of the Flattened Average Approach' has been provisionally accepted for publication in PLOS Computational Biology.

Best regards,

Leyla Isik

Associate Editor

PLOS Computational Biology

Samuel Gershman

Deputy Editor

PLOS Computational Biology

Reviewer's Responses to Questions

**Comments to the Authors:**

Reviewer #1: Dear authors,

Thank you for adding an informative simulation study investigating the sensitivity of their proposed method. This gives the reader an idea on how much is to be gained by using their proposed method instead of the standard in the field.

I spotted an annoying type:

Line 808-809 : “time-space position of the maximum power in the resulting volume”should be “time-frequency position of the maximum power in the resulting volume”.

**Have all data underlying the figures and results presented in the manuscript been provided?**

Reviewer #1: Yes

PLOS authors have the option to publish the peer review history of their article (what does this mean?). If published, this will include your full peer review and any attached files.

Reviewer #1: No

---

## [Editor Report · Acceptance letter]

2 Nov 2020

PCOMPBIOL-D-19-01389R2 

Breaking the Circularity in Circular Analyses: Simulations and Formal Treatment of the Flattened Average Approach

Dear Dr Bowman,

I am pleased to inform you that your manuscript has been formally accepted for publication in PLOS Computational Biology. Your manuscript is now with our production department and you will be notified of the publication date in due course.

With kind regards,

Matt Lyles
